# MINIFINETUNING: LOW-DATA GENERATION DOMAIN ADAPTATION THROUGH CORRECTIVE SELF-DISTILLATION

## ABSTRACT

Finetuning language models for a new domain inevitably leads to the deterioration of their general performance. This becomes more pronounced the more limited the finetuning data resource.

We introduce minifinetuning (MFT), a method for language model domain adaptation that considerably reduces the effects of overfitting-induced degeneralization in low-data settings and which does so in the absence of any pre-training data for replay. MFT demonstrates 2-10x more favourable specialization-to-degeneralization ratios than standard finetuning across a wide range of models and domains and exhibits an intrinsic robustness to overfitting when data in the new domain is scarce and down to as little as 500 samples.

Employing corrective self-distillation that is individualized on the sample level, MFT outperforms parameter-efficient finetuning methods, demonstrates replay-like forgetting mitigation properties, and is composable with either for a combined effect.

## 1 INTRODUCTION

Finetuning (FT) as a method for specializing models for new domains remains to be the dominant approach for reliable language model customization despite its relative maturity in the field (Liu et al., 2022). However, FT on a limited data budget can also lead to catastrophic forgetting that hinders model performance on the general domain (Kumar et al., 2022). This is illustrated in Figure 1, which plots specialization (improvement on the specialized domain) and degeneralization (detriment on the general domain) in terms of test perplexities for various data budgets throughout the process.

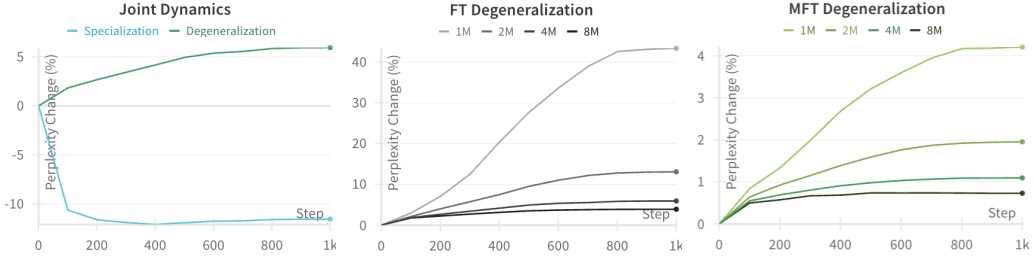

Figure 1: The specialization-degeneralization dynamics of domain adaptation finetuning computed as fractions of starting perplexity for various token budgets (1-8M) on PubMed corpus throughout 1K-step finetuning. *Left.* A joint visualization of specialization (% relative perplexity decrease on specialized domain) and degeneralization (% relative perplexity increase on the general domain) throughout a FT (4M) training instance. *Middle.* Degeneralization when using traditional hard-label finetuning. *Right.* Minifinetuning exhibits nine-fold lower levels of degeneralization consistently across all budgets.

There are two common remedies to the issue of finetuning-induced degeneralization. *Replay* stems from the classical literature on continual learning of neural networks (Sun et al., 2020; Peng et al., 2024; Shi et al., 2024) and consists of the re-introduction of some of the pre-training samples during FT. This requires access to pre-training (or equivalent) data and leads to a considerable increase in the compute budget. Moreover, generalist language models tend to come already finetuned for a wide range of tasks, and crude replay on the approximate distribution of the pre-training data risks tipping the carefully honed balance of changes introduced by post-training refinements (Dubey et al., 2024). *Parameter-efficient finetuning* (PEFT) is a more recent class of techniques conceived with the aim of reducing the computational and memory requirements for LLM finetuning (Houlsby et al., 2019; Hu et al., 2021; Liu et al., 2022; 2024). While the computational savings are the main benefit, PEFT methods also impose an often-tight constraint on the amount of representational power available for the model's adaptation to the new domain as a side effect. This acts as a natural backstop against model overfitting on the FT data (since the trainable representation capacity is insufficient to overwrite too much of the pre-trained knowledge), but being at a distance from the training process, it does not in any way guarantee that any amount of previous knowledge will be preserved. This can be fatally detrimental to generation, which in the longer span rests on entire sequences of tokens being predicted appropriately for the context. Marginal methods such as parameter ensembles and averaging or tailored adjustments to the optimization process also exist (Lin et al., 2024; Parmar et al., 2024).

Given the known shortfalls of the above remedies, it is desirable to have a technique that relies neither on modifications to the model nor on external data, but is largely independent and could be combined with them for an even better outcome. To this end, we introduce *minifinetuning* (MFT), which affects solely the finetuning training objective. At the heart of MFT is the construction of a per-token individualized, corrected distribution that combines the predicted distribution of the original model before finetuning and the ground truth token label of the finetuning data, but only to an extent that is not too destructive to the general knowledge of the model. Thoroughly ablating for every component of the distribution correction process, we find that "moving the goal posts" in the form of correcting the original unfinetuned model prediction towards the ground truth individually for every token is necessary for the model to improve by finetuning without significant detriment to the model's knowledge.

| Property | Method | | | | | |
|---|---|---|---|---|---|---|
| | FT | Replay | LoRA | DoRA | IA3 | MFT (ours) |
| specialization ($S$) | ★★★ | ★★☆ | ★⯪☆ | ★⯪☆ | ★☆☆ | ★★☆ |
| degeneralization ($DG$) | ★☆☆ | ★★★ | ★☆☆ | ★⯪☆ | ★★★ | ★★⯪ |
| designed to mitigate $DG$ | ✗ | ✓ | ✗ | ✗ | ✗ | ✓ |
| controllable $DG$-$S$ trade-off | ✗ | ✓ | ✓ | ✓ | ✗ | ✓ |
| original data not required | ✓ | ✗ | ✓ | ✓ | ✓ | ✓ |

Table 1: A qualitative comparison of available methods for LM low-data domain adaptation tuning. Replay (Sun et al., 2020), LoRA (Hu et al., 2021), DoRA (Liu et al., 2024), and IA3 (Liu et al., 2022) have been proposed by previous work.

MFT effectively amounts to algorithmically controlled adaptive self-distillation, in which a copy of the unfinetuned model acts as a teacher, the MFT corrective formula acts as an automated negotiator of teacher's knowledge with the incoming finetuning data, and the model under finetuning acts as a student (cf. Figure 4). In this sense, it is the teacher's predictions that act as the anchor preventing distant deviations from the pre-training data by communicating a compressed replacement for replay data in the form of soft labels.

We focus exclusively on generative language model adaptation for language generation on a new, specialized domain (e.g. a company knowledge base or literary works of one author), and do not examine cases in which the model undergoes FT for particular tasks or abilities (e.g. translation, summarization, reasoning). These cases, although too potential beneficiaries of minifinetuning, are not studied. Further, we consider domain adaptation finetuning to operate in low-data regime if the total data available is around or less than the amount of tokens in one batch used to pre-train the given model – usually in the order of millions for small- and medium-sized models – and firmly out of reach of in-context learning methods for such models. Such scenarios arise when tuning a

language model for a low-resource language (Cruz & Cheng, 2019), domain-specific terminology adaptation (Jeong, 2024), or language style transfer (Wang et al., 2019). In our experiments, we consider data budgets between 500 and 4000 full-text samples, corresponding to 1-8M tokens of text.

**Contributions.**

- We introduce MFT, a method for adapting models to new domains that exhibits markedly better trade-offs between forgetting of general domain and learning of specialized domain in low data settings (Section 2). MFT dwells in the modification of the training objective and is freely composable with existing dataset- and model-oriented methods that share this aim.

- We demonstrate the benefits of using MFT across a wide range of models and domains, evaluating MFT both in comparison to and in combination with replay and PEFT methods and showing 2-10x improvement in terms of the degeneralization-specialization trade-off (Section 3).

- We perform a thorough incremental ablation for each component of the MFT algorithm, demonstrating the necessity of the final formula (Section 4).

- We examine the response of FT/MFT finetuning to lowering data budgets, showing the eminent desirability of MFT over FT in low-data scenarios (Section 5).

The paper concludes with comparisons to related work and a discussion of limitations.

## 2 MINIFINETUNING

MFT consists of two components: (i) *a self-distillation teacher-student setup*; and (ii) *a distribution correction formula $DC\,(\cdot)$*.

**Self-distillation setup.** Before commencing training, an identical clone – the *teacher* – of the model to undergo MFT – the *student* – is created. One then proceeds to iterate through the provided data. For each batch, teacher/student forward passes are performed to find the teacher/student distributions. The teacher distribution is corrected according to the MFT distribution correction formula. The cross-entropy loss between the corrected distribution and the student distribution is computed, a backward pass is performed, and the weights are updated. See Algorithm 1. The training continues reusing data if needed until the termination by the user. This might be due to achieving the desired level of specialization or exceeding the maximum level of degeneralization permitted.

---

**Algorithm 1:** MFT training loop.

**Input:** dataset $\mathcal{D}$, frozen teacher model $T$, trainable student model $S$

**Output:** minifinetuned student model $S$

**for** *batch $\mathcal{B}$ from $\mathcal{D}$* **do**
$\quad p^T, p^S \leftarrow T\,(\mathcal{B})\,, S\,(\mathcal{B})$
$\quad p^C \leftarrow \mathrm{DC}\,(p^T)$

$\quad \ell \leftarrow \mathrm{CROSSENTROPY}\,(p^S, p^C)$
$\quad \mathrm{BACKWARD}\,(\ell, S)$
$\quad \mathrm{STEP}\,(S)$
**end**

---

**Distribution correction.** The distribution correction is performed individually for every token in every sample in the batch. Given a token position in a sample, let $l$ be the ground truth label (token vocabulary index), $p^T$ be the teacher distribution, and let $\mathbb{1}_i$ be the one-hot distribution concentrated at $i$. Denote the $i$-th element of a distribution $p$ by $p_i$.

If $\mathrm{argmax}\,p^T \neq l$, we want to make the minimal adjustment to the information of $p^T$ such that its most likely token becomes $l$ by some chosen target margin $\tau$. We formulate the "minimal adjustment" as follows: We want a new distribution $p^C$ whose argmax is $l$, whose argmax is separated in this new distribution from the previous (incorrect) argmax by a target correction $\tau$, and whose ratios of probabilities for every possible token pair except of $l$-pairs is preserved. This is easy to achieve: uniformly scale down the entire vector $p^T$ and add weight at $p_l^T$ so that $p_l^C$ is separated

from $p^T_{\text{argmax } p^T}$ by exactly $\tau$. Formally, we want

$$p^C = (1-\alpha)p^T + \alpha \mathbb{1}_l$$

where $\alpha$ is a scaling factor such that

$$(1-\alpha)p^T_l + \alpha 1 = (1-\alpha)p^T_{\text{argmax } p^T} + \tau.$$

Solving for $\alpha$, we get

$$\alpha = \frac{p^T_{\text{argmax } p^T} - p^T_l + \tau}{1 + p^T_{\text{argmax } p^T} - p^T_l}.$$

If argmax $p^T = l$, we want the probability of the token $l$ to improve as much as possible, but still be related to the previous (albeit correct) distribution. We therefore simply find a scaling factor $\beta$ such that $p^C_l = \min\left(1, p^T_l + \tau\right)$. This is at

$$\beta = \frac{\min\left(1, p^T_l + \tau\right) - p^T_l}{1 - p^T_l}$$

We define the distribution correction function

$$\text{DC}\left(p^T\right) = \begin{cases} (1-\alpha)p^T + \alpha \mathbb{1}_l & \text{if argmax } p^T \neq l \\ (1-\beta)p^T + \beta \mathbb{1}_l & \text{otherwise.} \end{cases}$$

Observe that if $\tau = 1$ then $\alpha, \beta = 1$ and this effectively reduces MFT to traditional finetuning. An illustration of the effect of the distribution correction is given in Figure 5, and the effect of $\tau$ is experimentally analyzed in Section 3.3 and Appendix F.

**Training cost.** In the most general setting, MFT requires two forward passes instead of one because of the addition of the teacher forward pass. Likewise, the memory required to store the model weights doubles when changing from FT to MFT. However, while twice the amount of memory is needed to store all parameters, the total training memory footprint is far from doubled. This is because the number of trainable parameters remains the same, and the main driver for the memory footprint is the optimizer state and is as much as 12 bytes (3-6x the parameter footprint) for Adam in mixed-precision (Rajbhandari et al., 2020). Furthermore, PEFT methods such as LoRA, DoRA, or IA3 (Hu et al., 2021; Liu et al., 2024; 2022) preserve original model parameters and thus duplicating model weights can be avoided when MFT is used, although two separate forward passes remain necessary.

## 3 EVALUATION

### 3.1 METHODOLOGY

We compare MFT to FT across a range of models on several domain-specialized datasets, and in combination with both replay and various PEFT methods.

**Models.** The general evaluation is performed on OpenELM 270M, OpenELM 450M, and OpenELM 1.1B (Mehta et al., 2024). These models all share the same tokenizer, the same pre-training recipe and datasets, and have undergone the same post-training adjustments. Keeping these factors constant allows us to examine the impact of model size on the FT/MFT effectiveness. Extended results for the GPT-Neo family (125M, 1.3B, 2.7B) (Black et al., 2021), Phi-1.5 and Phi-2 (1.3B, 2.7B) (Gunasekar et al., 2023; Abdin et al., 2024), Gemma 2B (Gemma Team et al., 2024), Minitron 4B (Muralidharan et al., 2024), and LLaMA 2 7B (Touvron et al., 2023) are listed in Appendices D and H.

**Data.** To test on three different specialized domains, we employ: (i) PMC Open Access Subset representing the medical domain (Bethesda, 2024); (ii) Pile of Law (Henderson et al., 2022); and (iii) OpenWebMath (Paster et al., 2023). To keep track of the model understanding of the general domain, we use OpenWebText (Gokaslan & Cohen, 2019), which we found not to result in any

significant distribution shift when preparing reference checkpoints (cf. Process). From each dataset we split off 0.5M-token worth of documents for validation and always use the same 4M-token worth of documents for training. The data frugality is intentional (cf. Section 1); note that the entire dataset is as big as a single batch of data used in LLaMa 2 pre-training (Touvron et al., 2023). We train on 2048-token sequences.

**Baselines.** Baseline FT is performed with no additional adjustments. When considering replay without further context, we design the experiments so that each batch contains 50% samples from the general domain and 50% samples from the specialized domain. From among the popular LM PEFT methods, we consider LoRA, DoRA, and IA3 (Hu et al., 2021; Liu et al., 2022; 2024), and where no further detail is given, the LoRA/DoRA rank is taken to be 8 and applied to all attention and MLP projections in the model.

**Process.** For each model, we first prepare a reference checkpoint lightly tuned on OpenWebText on at most 4M unique tokens and choose the checkpoint with the least validation perplexity. We note that this process often converges very quickly and before the data budget is reached, as the language models considered also tend to come from pre-training on general text (i.e. text not specific to any single domain). Then, for each model, each specialized-domain dataset, and each native/replay/PEFT method, we train two models, one utilizing classical FT and one with MFT. To report the results, we choose the checkpoint with least validation perplexity on the specialized domain. Each model is trained for 1000 steps with batch size 16, resulting in 32M tokens being seen during the course of training. With the training dataset fixed at 4M tokens, each context is seen 8 times on average by the end of the training, giving models ample opportunity to absorb the information and begin to overfit if prone to do so. For MFT training, we fix $\tau = 0.25$.

**Metrics.** We measure the relative decrease in validation perplexity on the specialized domain ("specialization", $S$), the relative increase in validation perplexity on the general domain ("degeneralization", $DG$), the ratio of the two relative changes in perplexity ($= DG/S$). Ideally, a good finetuning method would lead to high values of specialization $S$, low values of degeneralization $DG$, and by extension a low $DG/S$ ratio.

## 3.2 GENERAL EVALUATION

The results of the general evaluation carried according to the methodology set in Section 3.1 are given in Table 5 and elaborated on in Table 4. Extended results for more models are given in Table 6. As a general rule, all comparisons are performed with only the method changing and with all other things being kept equal. We observe a clear, regular pattern across all datasets and models that puts traditional FT as the winner in terms of the specialization ability but simultaneously shows MFT as the method leading to less degeneralization and much more favourable degeneralization-to-specialization ratios. We summarize our observations as follows:

**FT leads to more specialization than MFT.** Models trained using the traditional FT procedure show 25-35% higher levels of specialization over MFT in terms of relative perplexity improvement on specialized domains.

**MFT causes significantly less degeneralization than FT.** FT exhibits between three- and fifteen-fold increases in relative perplexity detriment on the general domain as a consequence. This is best captured by the $DG/S$ ratio, which is 2-4x more favourable for MFT across the setup.

**Replay excels at degeneralization mitigation.** Across the board, we observe a 5-12x reduction in degeneralization when using replay and when compared to FT. Recall, however, that replay has an unfair advantage over other contenders, as it has access to (pre-training) samples that other methods do not see.

**MFT outperforms LoRA and DoRA.** We see that across datasets and various model sizes, MFT outperforms LoRA and DoRA in terms of both higher specialization ability and lower degeneralization. Even where it performs roughly on-par in terms of specialization, it still exhibits lower levels

| Model | Method | Dataset | | | | | | | | |
|---|---|---|---|---|---|---|---|---|---|---|
| | | PubMed | | | Pile of Law | | | OpenWebMath | | |
| | | $S \uparrow$ | $DG \downarrow$ | ratio $\downarrow$ | $S \uparrow$ | $DG \downarrow$ | ratio $\downarrow$ | $S \uparrow$ | $DG \downarrow$ | ratio $\downarrow$ |
| OpenELM 270M | FT *(baseline)* | **10.9** | 1.0 | 0.09 | **14.5** | 1.2 | 0.08 | **2.7** | 0.5 | 0.19 |
| | LoRA | 8.5 | 1.7 | 0.20 | 10.5 | 1.5 | 0.14 | 2.1 | 0.9 | 0.45 |
| | DoRA | 8.5 | 1.7 | 0.19 | 10.5 | 1.4 | 0.14 | 2.1 | 0.9 | 0.44 |
| | IA3 | 0.9 | **0.0** | 0.03 | 1.3 | **0.1** | 0.07 | 0.3 | **0.0** | 0.06 |
| | MFT *(ours)* | 8.5 | 0.3 | **0.03** | 11.5 | 0.3 | **0.03** | 2.1 | 0.1 | **0.05** |
| | Replay | 8.6 | 0.1 | **0.01** | 11.7 | 0.1 | **0.01** | 2.2 | 0.1 | **0.04** |
| OpenELM 450M | FT *(baseline)* | **10.1** | 0.7 | 0.07 | **16.2** | 0.9 | 0.05 | **3.0** | 0.5 | 0.16 |
| | LoRA | 8.4 | 1.3 | 0.16 | 11.0 | 1.1 | 0.10 | 2.2 | 0.9 | 0.40 |
| | DoRA | 8.2 | 1.2 | 0.15 | 11.1 | 1.1 | 0.10 | 2.4 | 1.0 | 0.41 |
| | IA3 | 0.8 | **0.0** | 0.04 | 2.1 | **0.0** | 0.06 | 0.3 | **0.0** | 0.12 |
| | MFT *(ours)* | 8.4 | 0.3 | **0.04** | 12.9 | 0.3 | **0.02** | 2.3 | 0.2 | **0.07** |
| | Replay | 8.6 | 0.1 | **0.01** | 13.6 | 0.1 | **0.01** | 2.6 | 0.0 | **0.02** |
| OpenELM 1.1B | FT *(baseline)* | **9.3** | 0.7 | 0.07 | **16.9** | 0.9 | 0.05 | **3.5** | 0.5 | 0.14 |
| | LoRA | 8.2 | 1.6 | 0.20 | 9.8 | 1.3 | 0.14 | 2.3 | 0.8 | 0.35 |
| | DoRA | 8.2 | 1.6 | 0.20 | 9.8 | 1.3 | 0.13 | 2.3 | 0.8 | 0.36 |
| | IA3 | 0.6 | **0.0** | 0.05 | 1.3 | **0.0** | 0.08 | 0.3 | **0.0** | 0.16 |
| | MFT *(ours)* | 8.2 | 0.2 | **0.03** | 13.6 | 0.3 | **0.02** | 2.6 | 0.1 | **0.03** |
| | Replay | 8.4 | 0.1 | **0.01** | 14.5 | 0.1 | **0.01** | 3.0 | 0.0 | **0.00** |

Table 2: General evaluation of MFT ($\tau = 0.25$) across models of increasing size, various degeneralization mitigation techniques, and different specialized-domain datasets. $S$, $DG$, and $DG/S$ ratio are as in Section 3.1. $S, DG$ are reported in percentages, $DG/S$ is reported as a fraction. **Emphasis** marks the best value (highest for $S$; least for $DG$, ratio) for experiments that do not use general domain (pre-training) data, while **emphasis** marks the best value that was achieved only with the help of such data. Replay is further fenced out to highlight its access to pre-training data, in contrast to other methods. An extension of this table including results for method compositions is given in Appendix C. See the emphasized paragraphs of **??**

of degeneralization. An outlier to this trend on PEFT methods is IA3, which is conceptually different from LoRA/DoRA, injects much fewer learnable parameters into the model for training, and leads to both lower levels of specialization and degeneralization. We give a deeper analysis of the relationship between MFT and PEFT methods in Section 3.4.

**MFT can be composed with replay and PEFT methods.** The method composition results are given in Table 4. We observe that MFT can be composed with both replay and PEFT methods for a compound effect. This usually results in lower levels of specialization but also even lower levels of degeneralization. We note by composing two or more of these methods, one can leverage both the lower memory cost of PEFT methods and the lower degeneralization impact of finetuning due to replay and MFT.

The existence of the prevailing pattern and the overall favourability of the MFT ratios is well-aligned with the motivation and design goals of MFT, but it highlights the existence of a specialization-degeneralization trade-off: FT will lead to a model better adapted to the target low-resource specialized domain, but it will be at a cost of more degeneralization when compared to MFT. Fortunately, by the design of MFT, one can control the extent to which this trade-off applies by adjusting the target correction parameter $\tau$. We elaborate on this in Sections 3.3 and 3.4.

### 3.3 RELATIONSHIP TO REPLAY

In Section 2 we motivate the use of the unfinetuned model's logits as an "anchor" for the model to help it remain close to the general domain while specializing. The soft labels produced by the unfinetuned model act as highly compressed replay samples, where by providing the conditional

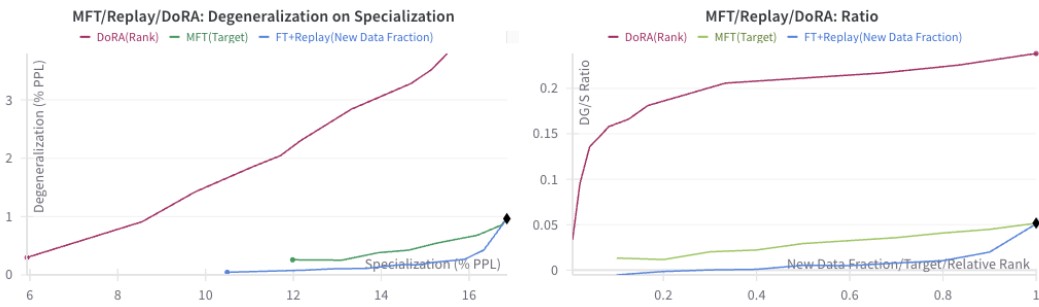

Figure 2: MFT of increasing target $\tau$ set against replay of increasing new data fraction $\nu$ and DoRA of increasing rank $\rho$. $\nu = \tau = 1$ corresponds to standard finetuning as is marked with ♦, $\rho = 1$ corresponds to DoRA rank 200. *Left.* Degeneralization plotted against specialization for different methods. DoRA displays clearly less favourable trade-off between $DG$ and $S$ across all ranks and when compared to all targets and new data fractions. Replay exhibits slightly better trade-off than MFT, but *requires the presence of general domain data*, which is not always available. *Right.* The $DG/S$ ratio plotted for increasing new data fraction, MFT target, and DoRA rank. Replay, MFT, and DoRA display the best ratios, in this order, where MFT and DoRA again compete at the disadvantage of not using general domain data.

probabilities for the next token one communicates a distilled next-token distribution of the general domain. This mitigates degeneralization without requiring access to the pre-training data.

Another similarity to replay offers itself in the form of the target correction parameter $\tau$. Just like the new data fraction $\nu \in [0, 1]$ controls the amount of new data from the specialized domain to be introduced in proportion to replay data from the general domain $\tau \in [0, 1]$ controls how much of the probability mass is to be reserved for the correction based on new data at the expense of the probability mass of the unfinetuned (teacher) model's token distribution soft labels. Observe that $\tau$ moves in the same direction as the new data fraction and that the replay fraction moves as $1 - \tau$.

We therefore run experiments for different values of $\nu$ and $\tau$ and measure their effect on specialization and degeneralization performance of the most specialized checkpoint. We follow the methodology of Section 3.1 but focus on OpenELM 1.1B and the Pile of Law dataset. The results of our experimentation are plotted in Figure 2. Additional experiments analyzing the effects of the new data fraction, MFT target $\tau$, and rank in detail were run in Appendices E to G, respectively.

We find that MFT of increasing target behaves in a vague similarity to FT aided by replay with increasing proportion of new data being shown to the model. While we observe that MFT leads to slightly higher levels of degeneralization, the difference is marginal (comp. Table 5), and this close adherence to replay behaviour is achieved in **total absence of replay data**. We conclude that the MFT mechanism of reusing soft labels of the unfinetuned teacher successfully helps to mimic the degeneralization mitigation of replay without relying on availability of the original training data.

## 3.4 RELATIONSHIP TO PEFT

In Section 1 we note that it is popular practice to use PEFT methods both for decreasing the memory/computational requirements of finetuning and for the mitigation of degeneralization, the latter appearing as a convenient side-effect of the reduction of the number of trainable parameters.

But how do PEFT methods fare against MFT in terms of specialization and degeneralization of the tuned models? We compare MFT to DoRA (Liu et al., 2024). DoRA has been recently shown to affect the weights of the resulting model less than the standard LoRA approach, meaning that it is positioned more favourably with respect to the design goals of MFT. The comparison is performed by running experiments for different values of rank (reported as relative rank $\rho = r/200$) and target $\tau$, and we measure the effect of the two methods in different configurations on specialization and degeneralization performance of the most-specialized checkpoint.

We follow the methodology of Section 3.1 but focus on OpenELM 1.1B and the Pile of Law dataset. The results of our experimentation are plotted in Figure 2. Detailed result listings and extended experimentation are reported in Appendix H.

We find that MFT outperforms DoRA for all ranks and sufficiently large target value $\tau$ in terms of specialization. Furthermore, for all ranks and targets, DoRA exhibits higher values of degeneralization than MFT, resulting in larger final $DG/S$ ratios. We conclude that MFT is consistently more robust to degeneralization than DoRA and outperforms DoRA in terms of specialization for sufficiently high values of $\tau$ while still causing lower levels of degeneralization. We therefore submit MFT as a replacement for DoRA in low-data scenarios.

## 4 ABLATION STUDY

MFT introduces a distinction between outputs whose argmax agrees with the token ground truth label (i.e. the correct predictions or correct prediction distributions) and those outputs for which this is not the case (incorrect predictions) and that thus require a correction (cf. Section 2). With this dichotomy in mind, we can consider an incremental sequence of four relevant methods.

*Finetuning.* This is the standard finetuning approach. The distribution of the finetuned model is trained against the one-hot target distribution of the ground truth regardless of whether the unfinetuned model prediction distribution for a given token is correct with respect to the ground truth.

*Distillation & corrective finetuning on incorrect tokens.* One can require that the model keeps its original (unfinetuned) distribution on inputs where it produces the correct prediction distributions, and learns the ground truth label only on those tokens where it produces incorrect prediction distributions. This can be realized by training the model to distill the unfinetuned model on the correct tokens and learn the one-hot distributions on inputs leading to incorrect predictions as in FT. In this manner, the prediction distributions of the unfinetuned model are to serve as a form of anchoring preventing a destructive distribution shift from the original to the new domain.

*Distillation & corrective distillation on incorrect tokens.* Moving one step further on the above method, one can insist that the incorrect distributions are first corrected and then passed to the model in place of the ground truth one-hot distributions. This behaviour can be achieved by fixing $\beta = 0$ in the distribution correction formula $DC\left(p^T\right)$ (cf. Section 2). With the corrections to the training target affecting only the targets where the unfinetuned model predictions are incorrect, one can speak of singly-corrective distillation.

*Corrective distillation on both correct and incorrect tokens – Minifinetuning.* This is our method, outlined in detail in Section 2. Given that the target distributions for both the correct and incorrect predictions is threshold-adjusted (leading to the scaling factors $\alpha, \beta$ in $DC\left(p^T\right)$), we may think of MFT as of doubly-corrective distillation.

We perform incremental ablation, which includes the measurement of performance of the two methods more complex than the baseline FT but less complex than MFT. We mirror the setup of Section 3 but restrict our ablation to the Pile of Law and smaller models of the OpenELM family. The results are listed in Table 3, and we make a number of observations.

| Method | Model | | | | | | | | |
|---|---|---|---|---|---|---|---|---|---|
| | OpenELM 270M | | | OpenELM 450M | | | OpenELM 1.1B | | |
| | $S \uparrow$ | $DG \downarrow$ | ratio $\downarrow$ | $S \uparrow$ | $DG \downarrow$ | ratio $\downarrow$ | $S \uparrow$ | $DG \downarrow$ | ratio $\downarrow$ |
| finetuning (FT) | **14.5** | 1.2 | 0.08 | **16.2** | 0.9 | 0.05 | **16.9** | 0.9 | 0.05 |
| + distillation | 10.1 | 1.4 | 0.14 | 10.0 | 1.1 | 0.11 | 10.8 | 1.1 | 0.10 |
| + partial correction | 9.7 | 0.5 | 0.05 | 11.1 | 0.4 | 0.04 | 11.9 | 0.5 | 0.04 |
| + full correction (MFT) | 11.5 | **0.3** | **0.03** | 12.9 | **0.3** | **0.02** | 13.6 | **0.3** | **0.02** |

Table 3: Incremental ablation of MFT; metrics computed as in Table 5. **Emphasis** marks the best overall performance. See the emphasized paragraphs of Section 4 for interpretation.

**MFT outperforms its peers.**    Consistently with Table 5, we find that FT performs the best in terms of the relative specialized-domain perplexity improvement, and MFT performs the best in terms of the relative general-domain perplexity detriment and the $DG/S$ ratio.

**Corrective finetuning hurts models the most.**    We observe that the naive combination of distillation on correct predictions and standard finetuning on incorrect predictions leads to even worse ratio performance than the standard FT alone, exhibiting smaller specialization than both FT and MFT and worse degeneralization than FT. The $DG/S$ ratios for this method are the smallest among all the methods.

**Single correction is not enough.**    We find that the combination of untouched distillation and corrective distillation on incorrect predictions performs better than corrective finetuning but still worse than MFT.

In sum, the poor performance of corrective finetuning justifies the inclusion of some form of distribution correction to MFT, and the worse-than-MFT performance of singly-corrective distillation justifies the double correction, i.e. the correction of both the correct and incorrect model predictions as seen in MFT.

## 5  RESPONSE TO DATA SCARCITY

We examine and compare the responses of FT and MFT to falling data budgets by tracking the levels of specialization, degeneralization, and degeneralization-specialization ratio throughout training instances of each methods. For setup, we follow the recipe in Section 3.1 and narrow the scope to the response of OpenELM 270M on PMC Open Access dataset. All metrics are measured on appropriate validation splits. The experimentation is visualized in Figure 3.

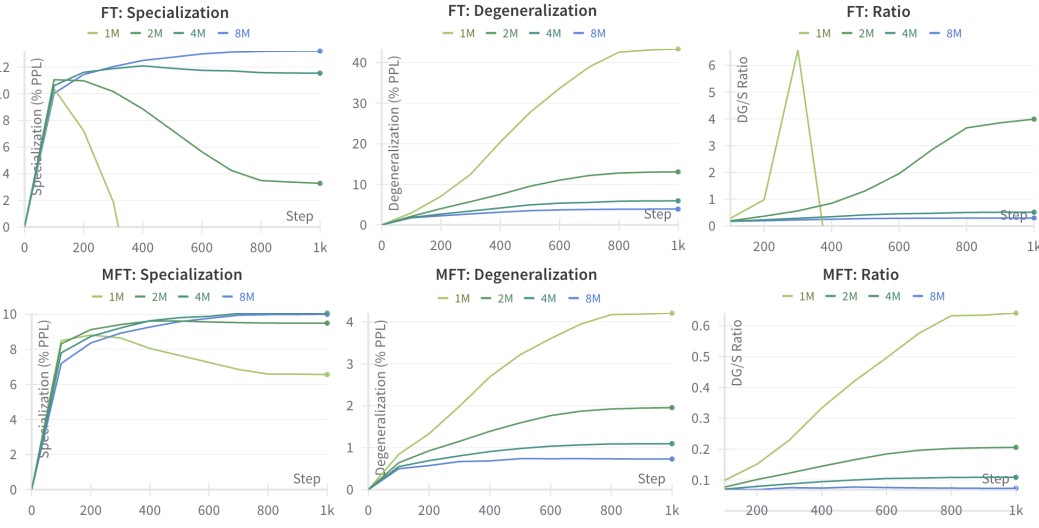

Figure 3: *Top/Bottom.* Response of FT/MFT to different level of data scarcity. 1M-8M token budgets correspond to 500-4000 sample budgets as per the setup of Section 3.1.

On the outset, we observe that MFT and FT both exhibit decreases of perplexity on the specialized domain in the same order of magnitude but differ by a full order of magnitude in the increases in perplexity on the general domain. Inspecting the individual series for different budgets, we observe that FT results in overfitting on the specialized domain much more readily than MFT. In the case of 1M-token data budget, FT even results in worse-than-initial performance on the specialized domain after just a few hundred steps. Meanwhile, MFT appears to be naturally restrained from such excessive overfitting and manages to keep some level of improvement on the specialized domain by the end of the training for all data budgets.

In sum, MFT demonstrates much higher robustness to destructive levels of overfitting throughout the training in comparison to FT while delivering slightly weaker specialization.

## 6 RELATED WORK

**Domain adaptation.** Previous work on language model domain adaptation recognizes the inherent parameter-sharing behaviour (Chronopoulou et al., 2021) and domain-mixing representation entanglement (Li et al., 2020) arising from continued model training on a specialized domain. These are then made use of by separating domain-specific parameters and decomposing representations, respectively, or countered by adversarial training objective adjustments (Vu et al., 2020) in order to achieve greater inference efficiency or predictive performance. All of this work, however, implicitly assumes that a sufficient data mass is available for the new domain and does not concern itself with the severe consequences of tuning when data is scarce.

**Low-resource domain adaptation.** Recent studies addressing the problem of low-resource domain adaptation in language modeling (Diao et al., 2021; Huang et al., 2023) make use of full-scale meta-models wrapping around the original language models in order to aid their language understanding or generation abilities. This is in contrast with our method, which still focuses on adapting the original model as a monolith. Moreover, the definition of a "low-resource" setting varies, with Diao et al. (2021); Chen et al. (2023) considering order(s) of magnitude larger data budgets than ours to already fall into this category, even though the methods might themselves resort to data filtering to bolster efficiency gains. To the best of our knowledge, no previous work examines the problem of model adaptation through directly tuning on scarce data.

**Parameter-efficient FT.** As detailed in Section 1, PEFT methods have been observed to act as a natural backstop to model overfitting due to the limited representational power they lend to the process of finetuning. This property has seen use in low-resource settings (Su et al., 2024; Zhang et al., 2024), though the studies generally observe varying degrees of success across different and differently-configured PEFT methods. In contrast, MFT behaves consistently across entire families of models (cf. Section 3.2) and applications of PEFT methods (cf. Section 3.4), and acts as a flexible complementary technique.

**Self-distillative FT.** A recent work proposes to leverage hard-label self-distillation for instruction data augmentation in order to reduce distribution shift introduced by post-training instruction tuning (Yang et al., 2024). While closely related to MFT in its goals, the work takes an ahead-of-time generative data-augmentation approach rather than the on-the-fly soft-label distillation approach taken by us, and is tailored to work on chat-style instruction datasets that are an order of magnitude larger.

## 7 LIMITATIONS

**Model tuning for instruction following.** We do not adjust for nor test our method in its present form on instruction tuning datasets. This is because the most effective instruction tuning post-training routines already come carefully configured and tend to operate on larger blends of post-training data that make them fall outside the category of low-resource FT (Gemma Team et al., 2024; Dubey et al., 2024). Nevertheless, we recognize this direction as a natural avenue for a broader application of our method.

**Transferring task performance to specialized domains.** Conversely to the above point, one limitation of this study is that it does not examine the effect MFT domain adaptation for language generation has on model's performance on general tasks. Our primary adaptation of all models to general domain represented by one dataset in order to provide a universal starting point for the assessment of specialization and degeneralization is an obstacle to such evaluation as it makes the models forget parts of the knowledge gained in their post-training even before they begin to adapt to the specialized domain.

**Document-level tuning.** A natural extension of Section 5 would be to examine the effect of MFT vs that of FT on single-document finetuning. We acknowledge this as another limitation of our study but note that such experimentation is not necessary for a convincing demonstration of MFT's prowess in low-data scenarios.

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

# A MFT SETUP DIAGRAM

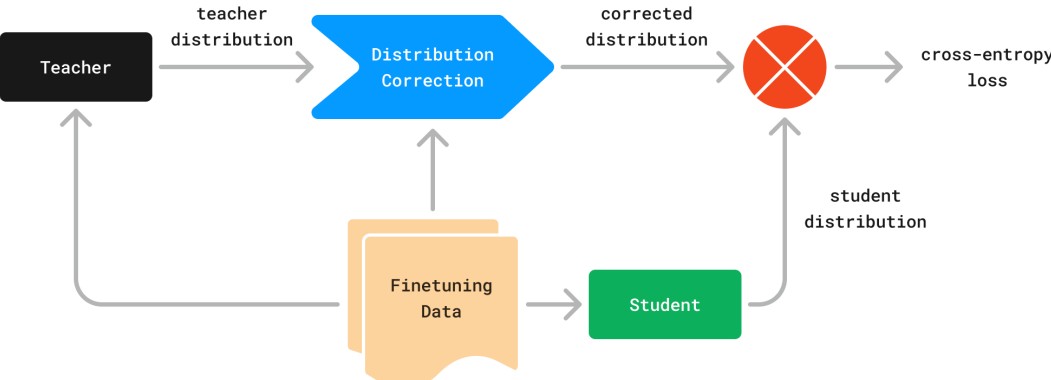

Figure 4: The MFT setup as explained in Section 2. The student model (bottom) trains to match corrected soft labels (top-right) of its own unfinetuned predictions produced by the teacher (top-left). Observe that only finetuning data is used (meaning that pre-training general domain data is not necessary), and that the teacher's predictions are customized on a per-token basis to by appropriately $\tau$-corrected for the student's learning.

# B AN ILLUSTRATION OF THE DISTRIBUTION CORRECTION

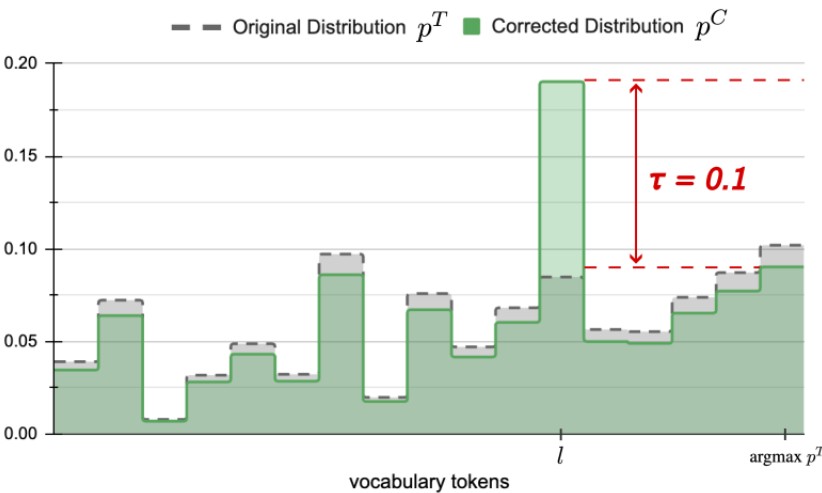

Figure 5: The MFT distribution correction as formulated in Section 2. The original distribution is uniformly scaled down to release probability mass for an offsetting correction by threshold $\tau$. Observe that the original distribution provided by the teacher (gray) is uniformly ($\alpha$-)scaled down to produce the corrected distribution (green) on all tokens but $l$ (the ground truth token), where the released probability mass is added to enforce target $\tau$ separation of $p_l^C$ from $p_{\mathrm{argmax}\, p^T}^C$.

## C  GENERAL EVALUATION INCLUDING COMPOSITIONS

| Model | Method | PubMed | | | Pile of Law | | | OpenWebMath | | |
|---|---|---|---|---|---|---|---|---|---|---|
| | | $S\uparrow$ | $DG\downarrow$ | ratio $\downarrow$ | $S\uparrow$ | $DG\downarrow$ | ratio $\downarrow$ | $S\uparrow$ | $DG\downarrow$ | ratio $\downarrow$ |
| OpenELM 270M | FT *(baseline)* | **10.9** | 1.0 | 0.09 | **14.5** | 1.2 | 0.08 | **2.7** | 0.5 | 0.19 |
| | MFT *(ours)* | 8.5 | **0.3** | **0.03** | 11.5 | **0.3** | **0.03** | 2.1 | **0.1** | **0.05** |
| | Replay | **8.6** | 0.1 | 0.01 | **11.7** | 0.1 | 0.01 | **2.2** | 0.1 | 0.04 |
| | Replay+MFT | 6.7 | **0.0** | **0.00** | 9.4 | **0.0** | **0.00** | 1.7 | **0.0** | **0.01** |
| | LoRA | **8.5** | 1.7 | 0.20 | **10.5** | 1.5 | 0.14 | **2.1** | 0.9 | 0.45 |
| | LoRA+MFT | 6.7 | **0.4** | **0.06** | 10.2 | **0.5** | **0.05** | 1.8 | **0.2** | **0.13** |
| | DoRA | **8.5** | 1.7 | 0.19 | **10.5** | 1.4 | 0.14 | **2.1** | 0.9 | 0.44 |
| | DoRA+MFT | 6.7 | **0.4** | **0.06** | 10.2 | **0.5** | **0.05** | 1.8 | **0.2** | **0.14** |
| | IA3 | **0.9** | 0.0 | 0.03 | **1.3** | 0.1 | 0.07 | **0.3** | 0.0 | 0.06 |
| | IA3+MFT | 0.6 | **0.0** | **0.00** | **1.3** | **0.0** | **0.03** | 0.3 | **0.0** | **0.04** |
| OpenELM 450M | FT *(baseline)* | **10.1** | 0.7 | 0.07 | **16.2** | 0.9 | 0.05 | **3.0** | 0.5 | 0.16 |
| | MFT *(ours)* | 8.4 | **0.3** | **0.04** | 12.9 | **0.3** | **0.02** | 2.3 | **0.2** | **0.07** |
| | Replay | **8.6** | 0.1 | 0.01 | **13.6** | 0.1 | 0.01 | **2.6** | 0.0 | 0.02 |
| | Replay+MFT | 6.5 | **0.0** | **0.00** | 11.0 | **0.0** | **0.00** | 1.8 | **0.0** | **0.01** |
| | LoRA | **8.4** | 1.3 | 0.16 | **11.0** | 1.1 | 0.10 | **2.2** | 0.9 | 0.40 |
| | LoRA+MFT | 6.6 | **0.4** | **0.06** | 10.4 | **0.5** | **0.05** | 1.9 | **0.2** | **0.09** |
| | DoRA | **8.2** | 1.2 | 0.15 | **11.1** | 1.1 | 0.10 | **2.4** | 1.0 | 0.41 |
| | DoRA+MFT | 6.5 | **0.4** | **0.06** | 10.4 | **0.4** | **0.04** | 1.8 | **0.2** | **0.09** |
| | IA3 | **0.8** | 0.0 | 0.04 | **2.1** | 0.0 | 0.06 | **0.3** | 0.0 | 0.12 |
| | IA3+MFT | 0.7 | **0.0** | **0.03** | 1.7 | **0.0** | **0.03** | 0.3 | **0.0** | **0.08** |
| OpenELM 1.1B | FT *(baseline)* | **9.3** | 0.7 | 0.07 | **16.9** | 0.9 | 0.05 | **3.5** | 0.5 | 0.14 |
| | MFT *(ours)* | 8.2 | **0.2** | **0.03** | 13.6 | **0.3** | **0.02** | 2.6 | **0.1** | **0.03** |
| | Replay | **8.4** | 0.1 | 0.01 | **14.5** | 0.1 | 0.01 | **3.0** | 0.0 | 0.00 |
| | Replay+MFT | 6.6 | **0.0** | **0.00** | 11.7 | **0.0** | **0.00** | 2.0 | **0.0** | **0.00** |
| | LoRA | **8.2** | 1.6 | 0.20 | **9.8** | 1.3 | 0.14 | **2.3** | 0.8 | 0.35 |
| | LoRA+MFT | 6.4 | **0.3** | **0.05** | 9.1 | **0.4** | **0.04** | 1.8 | **0.2** | **0.11** |
| | DoRA | **8.2** | 1.6 | 0.20 | **9.8** | 1.3 | 0.13 | **2.3** | 0.8 | 0.36 |
| | DoRA+MFT | 6.4 | **0.3** | **0.05** | 9.2 | **0.4** | **0.04** | 1.8 | **0.2** | **0.12** |
| | IA3 | **0.6** | 0.0 | 0.05 | 1.3 | 0.0 | 0.08 | **0.3** | 0.0 | 0.16 |
| | IA3+MFT | 0.6 | **0.0** | **0.01** | **1.4** | **0.0** | **0.00** | 0.3 | **0.0** | **0.07** |

Table 4: General evaluation of MFT ($\tau = 0.25$) across models of increasing size, various degeneralization mitigation techniques, and different specialized-domain datasets. $S$, $DG$, and $DG/S$ ratio are as in Section 3.1. $S$, $DG$ are reported in percentages, $DG/S$ is reported as a fraction. **Emphasis** marks the better value (greater for $S$; smaller for $DG$, ratio) for each pair of experiments.

## D EXTENDED GENERAL EVALUATION

| Model | Method | Dataset | | | | | | | | |
|---|---|---|---|---|---|---|---|---|---|---|
| | | PubMed | | | Pile of Law | | | OpenWebMath | | |
| | | $S \uparrow$ | $DG \downarrow$ | ratio $\downarrow$ | $S \uparrow$ | $DG \downarrow$ | ratio $\downarrow$ | $S \uparrow$ | $DG \downarrow$ | ratio $\downarrow$ |
| Llama 3.1 8B | FT | 7.0 | 1.4 | 0.20 | 6.4 | 1.5 | 0.23 | 2.6 | 0.9 | 0.35 |
| | LoRA | 6.8 | 1.8 | 0.26 | 5.2 | 1.2 | 0.23 | 2.8 | 0.7 | 0.25 |
| | DoRA | 6.9 | 1.6 | 0.23 | 5.0 | 1.1 | 0.22 | 2.8 | 0.9 | 0.26 |
| | IA3 | 1.2 | 0.1 | 0.08 | 1.5 | 0.0 | 0.01 | 0.6 | 0.0 | 0.08 |
| | MFT | 6.6 | 0.9 | 0.14 | 5.1 | 0.4 | 0.08 | 2.3 | 0.4 | 0.17 |
| | Replay | 6.4 | 0.1 | 0.02 | 5.4 | 0.2 | 0.04 | 2.3 | 0.2 | 0.09 |
| | EWC | 6.2 | 1.2 | 0.19 | 5.3 | 1.1 | 0.21 | 2.6 | 1.0 | 0.38 |
| Llama 3.2 1B | FT | 11.2 | 3.5 | 0.31 | 12.0 | 5.2 | 0.43 | 2.1 | 1.0 | 0.48 |
| | LoRA | 10.8 | 3.8 | 0.35 | 9.9 | 5.0 | 0.51 | 2.0 | 0.9 | 0.45 |
| | DoRA | 10.8 | 3.2 | 0.30 | 9.9 | 5.0 | 0.51 | 1.9 | 0.8 | 0.42 |
| | IA3 | 2.3 | 0.4 | 0.17 | 1.7 | 0.1 | 0.06 | 0.1 | 0.0 | 0.04 |
| | MFT | 10.5 | 1.7 | 0.16 | 9.7 | 1.5 | 0.15 | 1.6 | 0.3 | 0.19 |
| | Replay | 7.5 | 1.5 | 0.20 | 11.3 | 1.0 | 0.09 | 2.1 | 0.0 | 0.02 |
| | EWC | 7.9 | 2.6 | 0.33 | 11.0 | 4.9 | 0.45 | 2.3 | 1.0 | 0.43 |
| Llama 3.2 3B | FT | 9.1 | 1.6 | 0.18 | 8.3 | 2.8 | 0.34 | 4.0 | 1.9 | 0.48 |
| | LoRA | 8.2 | 1.6 | 0.20 | 7.8 | 2.6 | 0.33 | 3.3 | 1.7 | 0.52 |
| | DoRA | 8.2 | 1.5 | 0.18 | 7.8 | 2.6 | 0.33 | 3.3 | 1.7 | 0.52 |
| | IA3 | 2.1 | 0.2 | 0.10 | 1.7 | 0.1 | 0.06 | 0.3 | 0.1 | 0.33 |
| | MFT | 7.6 | 0.9 | 0.12 | 7.5 | 0.8 | 0.11 | 3.1 | 0.4 | 0.13 |
| | Replay | 6.8 | 0.4 | 0.06 | 7.9 | 0.6 | 0.08 | 3.3 | 0.1 | 0.03 |
| | EWC | 7.2 | 1.2 | 0.17 | 7.9 | 2.2 | 0.28 | 3.9 | 1.7 | 0.44 |

Table 5: Evaluation of MFT ($\tau = 0.25$) across most recent models, various degeneralization mitigation techniques, and different specialized-domain datasets. $S$, $DG$, and $DG/S$ ratio are as in Section 3.1. $S, DG$ are reported in percentages, $DG/S$ is reported as a fraction. Replay and the Elastic Weight Consolidation (EWC, Kirkpatrick et al. (2017)) are further fenced out to highlight their access to pre-training data (or a proxy thereof, namely the Fisher scores), in contrast to other methods.

| Model | Method | Dataset | | | | | | | | |
|---|---|---|---|---|---|---|---|---|---|---|
| | | PubMed | | | Pile of Law | | | OpenWebMath | | |
| | | $S\uparrow$ | $DG\downarrow$ | ratio $\downarrow$ | $S\uparrow$ | $DG\downarrow$ | ratio $\downarrow$ | $S\uparrow$ | $DG\downarrow$ | ratio $\downarrow$ |
| GPT Neo 2.7B | FT | **9.8** | 0.9 | 0.09 | **8.8** | 1.5 | 0.17 | **2.0** | 0.7 | 0.34 |
| | MFT | 7.0 | **0.2** | **0.03** | 6.6 | **0.4** | **0.06** | 1.4 | **0.2** | **0.15** |
| | Replay | **5.5** | 0.2 | **0.04** | **6.3** | 0.2 | 0.03 | **1.5** | 0.2 | **0.12** |
| | Replay+MFT | 3.8 | **0.1** | **0.04** | 4.7 | **0.1** | **0.02** | 1.0 | **0.1** | 0.13 |
| | LoRA | **9.7** | 1.1 | 0.11 | **6.8** | 2.3 | 0.34 | **1.9** | 1.1 | 0.60 |
| | LoRA+MFT | 6.5 | **0.3** | **0.04** | 4.6 | **0.3** | **0.07** | 1.2 | **0.2** | **0.19** |
| | DoRA | **9.6** | 1.2 | 0.12 | **6.7** | 2.4 | 0.35 | **2.0** | 1.2 | 0.60 |
| | DoRA+MFT | 6.6 | **0.2** | **0.04** | 4.9 | **0.4** | **0.08** | 1.3 | **0.2** | **0.18** |
| Phi 1.5 | FT | **16.8** | 1.8 | 0.11 | **28.4** | 5.1 | 0.18 | **3.9** | 1.4 | 0.36 |
| | MFT | 11.0 | **0.4** | **0.03** | 20.4 | **0.8** | **0.04** | 2.9 | **0.4** | **0.12** |
| | Replay | **10.9** | 0.2 | 0.02 | **22.8** | 0.5 | 0.02 | **3.2** | 0.2 | 0.06 |
| | Replay+MFT | 6.7 | **0.0** | **0.01** | 16.5 | **0.1** | **0.00** | 2.2 | **0.0** | **0.01** |
| | LoRA | **14.0** | 2.0 | 0.15 | **20.6** | 6.7 | 0.32 | **3.1** | 1.4 | 0.46 |
| | LoRA+MFT | 8.8 | **0.4** | **0.04** | 14.5 | **1.1** | **0.07** | 2.1 | **0.3** | **0.15** |
| | DoRA | **14.0** | 2.0 | 0.14 | **20.6** | 6.6 | 0.32 | **3.1** | 1.4 | 0.46 |
| | DoRA+MFT | 8.9 | **0.4** | **0.04** | 14.3 | **1.0** | **0.07** | 2.1 | **0.3** | **0.14** |
| Phi 2 | FT | **7.5** | 1.4 | 0.19 | **12.5** | 2.4 | 0.19 | **1.6** | 1.0 | 0.60 |
| | MFT | 5.5 | **0.6** | **0.10** | 9.7 | **1.0** | **0.10** | 1.3 | **0.5** | **0.40** |
| | Replay | **4.0** | 0.5 | 0.11 | **9.3** | 0.5 | 0.05 | **1.1** | 0.6 | 0.54 |
| | Replay+MFT | 3.0 | **0.2** | **0.06** | 7.3 | **0.3** | **0.03** | 1.1 | **0.2** | **0.15** |
| | LoRA | **3.8** | 0.9 | 0.23 | **5.4** | 2.8 | 0.52 | **1.3** | 0.7 | 0.54 |
| | LoRA+MFT | 2.2 | **0.3** | **0.16** | 4.0 | **0.8** | **0.20** | 0.9 | **0.2** | **0.25** |
| | DoRA | **3.7** | 0.9 | 0.25 | **5.5** | 2.6 | 0.47 | **1.3** | 0.8 | 0.60 |
| | DoRA+MFT | 2.4 | **0.4** | **0.15** | 4.0 | **0.7** | **0.19** | 0.8 | **0.3** | **0.33** |

Table 6: Evaluation of MFT against FT when used in conjunction with LoRA/DoRA across additional smaller models (Phi 1.5, Phi 2, and GPT Neo 2.7B) and several specialized-domain datasets; metrics computed as in Section 3.1. $S, DG$ are reported in percentages, $DG/S$ is reported as a fraction. **Emphasis** marks the better value.

# E   DETAILED REPLAY PERFORMANCE

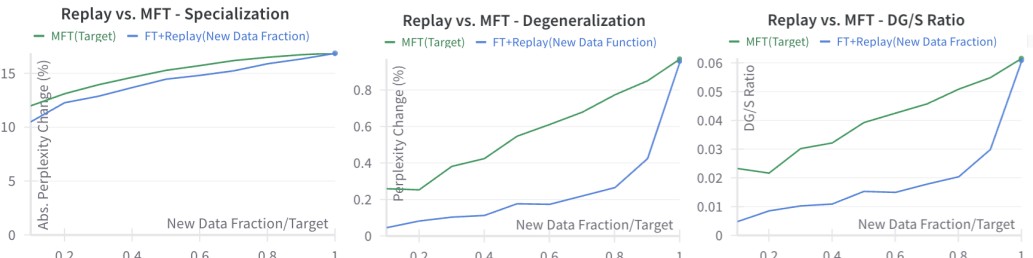

Figure 6: Replay of increasing new data fraction set against MFT of increasing target on the Pile of Law dataset. $\nu = \tau = 1$ corresponds to standard finetuning. *Left.* MFT demonstrates slightly better (1-2 ppts) specialization than FT+Replay. *Middle.* MFT exhibits slightly higher (0.2-0.4 ppts) degeneralization than FT+Replay. *Right.* As a result, MFT exhibits 20-60% higher $DG/S$ ratios than FT+Replay.

| $\nu$ | Model | | | | | | | | |
|---|---|---|---|---|---|---|---|---|---|
| | OpenELM 270M | | | OpenELM 450M | | | OpenELM 1.1B | | |
| | $S\uparrow$ | $DG\downarrow$ | ratio $\downarrow$ | $S\uparrow$ | $DG\downarrow$ | ratio $\downarrow$ | $S\uparrow$ | $DG\downarrow$ | ratio $\downarrow$ |
| 100% | **14.5** | 1.2 | 0.08 | **16.2** | 0.9 | 0.06 | **16.9** | 0.9 | 0.05 |
| 90% | 14.0 | 0.6 | 0.05 | 15.5 | 0.4 | 0.03 | 16.3 | 0.3 | 0.02 |
| 80% | 13.5 | 0.4 | 0.03 | 15.3 | 0.3 | 0.02 | 15.9 | 0.2 | 0.01 |
| 70% | 13.0 | 0.3 | 0.02 | 14.8 | 0.2 | 0.01 | 15.2 | 0.1 | 0.01 |
| 60% | 12.5 | 0.2 | 0.01 | 14.2 | 0.2 | 0.01 | 14.8 | 0.1 | 0.00 |
| 50% | 11.7 | 0.1 | 0.01 | 13.6 | 0.1 | 0.01 | 14.5 | 0.1 | 0.01 |
| 40% | 11.1 | 0.1 | 0.01 | 13.1 | 0.1 | 0.00 | 13.7 | 0.0 | 0.00 |
| 30% | 10.3 | 0.0 | 0.00 | 12.2 | 0.0 | 0.00 | 12.9 | 0.0 | 0.00 |
| 20% | 9.3 | 0.0 | 0.00 | 11.3 | 0.0 | 0.00 | 12.3 | 0.0 | 0.00 |
| 10% | 7.7 | **0.0** | **0.00** | 9.6 | **0.0** | **0.00** | 10.5 | **0.0** | **0.00** |

Table 7: Detailed replay results for FT on the Pile of Law dataset; metrics computed as in Table 5. **Emphasis** marks the best overall performance. $\nu = 100\%$ corresponds to plain FT, $\nu = 0\%$ means no new data is being introduced and so the model is not being tuned.

# F  DETAILED MFT TARGET DEPENDENCE

| $\tau$ | Model | | | | | | | | |
|---|---|---|---|---|---|---|---|---|---|
| | OpenELM 270M | | | OpenELM 450M | | | OpenELM 1.1B | | |
| | $S \uparrow$ | $DG \downarrow$ | ratio $\downarrow$ | $S \uparrow$ | $DG \downarrow$ | ratio $\downarrow$ | $S \uparrow$ | $DG \downarrow$ | ratio $\downarrow$ |
| 1.0 (FT) | **14.5** | 1.2 | 0.08 | **16.2** | 0.9 | 0.06 | **16.9** | 0.9 | 0.05 |
| 0.9 | 14.4 | 1.1 | 0.08 | 15.8 | 0.8 | 0.05 | 16.7 | 0.8 | 0.04 |
| 0.8 | 14.2 | 0.9 | 0.07 | 15.8 | 0.7 | 0.05 | 16.5 | 0.7 | 0.04 |
| 0.7 | 13.9 | 0.9 | 0.06 | 15.3 | 0.6 | 0.04 | 16.2 | 0.6 | 0.04 |
| 0.6 | 13.5 | 0.8 | 0.06 | 15.1 | 0.6 | 0.04 | 15.7 | 0.5 | 0.03 |
| 0.5 | 13.1 | 0.7 | 0.05 | 14.6 | 0.6 | 0.04 | 15.3 | 0.4 | 0.03 |
| 0.4 | 12.6 | 0.5 | 0.04 | 14.0 | 0.4 | 0.03 | 14.6 | 0.3 | 0.02 |
| 0.3 | 11.9 | 0.4 | 0.04 | 13.3 | 0.3 | 0.03 | 13.9 | 0.3 | 0.02 |
| 0.2 | 11.1 | 0.2 | 0.02 | 12.3 | 0.2 | 0.02 | 13.1 | 0.2 | 0.01 |
| 0.1 | 9.9 | 0.2 | 0.02 | 11.2 | 0.2 | 0.02 | 12.0 | 0.2 | 0.01 |
| 0.0 | 8.0 | **0.2** | **0.02** | 9.1 | **0.2** | **0.02** | 9.8 | **0.1** | **0.01** |

Table 8: Detailed target dependence results for MFT on the Pile of Law dataset; metrics computed as in Table 5. **Emphasis** marks the best overall performance. $\tau = 1$ corresponds to plain FT.

# G   DETAILED DORA RANK DEPENDENCE

| $r$ | Model | | | | | | | | |
|---|---|---|---|---|---|---|---|---|---|
| | OpenELM 270M | | | OpenELM 450M | | | OpenELM 1.1B | | |
| | $S\uparrow$ | $DG\downarrow$ | ratio $\downarrow$ | $S\uparrow$ | $DG\downarrow$ | ratio $\downarrow$ | $S\uparrow$ | $DG\downarrow$ | ratio $\downarrow$ |
| 1 | 5.5 | **0.4** | **0.06** | 6.6 | **0.1** | **0.02** | 5.9 | **0.2** | **0.03** |
| 4 | 8.8 | 0.9 | 0.10 | 9.6 | 0.7 | 0.07 | 8.6 | 0.8 | 0.09 |
| 8 | 10.5 | 1.5 | 0.14 | 11.0 | 1.1 | 0.10 | 9.8 | 1.3 | 0.14 |
| 16 | 12.2 | 2.1 | 0.17 | 12.6 | 1.8 | 0.15 | 11.0 | 1.7 | 0.16 |
| 24 | 13.4 | 2.5 | 0.18 | 13.5 | 2.4 | 0.18 | 11.7 | 1.9 | 0.17 |
| 32 | 13.9 | 2.8 | 0.20 | 14.3 | 2.5 | 0.18 | 12.2 | 2.2 | 0.18 |
| 64 | 16.1 | 3.3 | 0.21 | 15.9 | 3.3 | 0.21 | 13.3 | 2.7 | 0.21 |
| 128 | 18.3 | 4.1 | 0.22 | 17.9 | 3.9 | 0.22 | 14.7 | 3.2 | 0.22 |
| 160 | 19.2 | 4.3 | 0.23 | 18.6 | 4.2 | 0.23 | 15.1 | 3.4 | 0.23 |
| 192 | 19.9 | 4.6 | 0.23 | 19.3 | 4.5 | 0.23 | 15.5 | 3.7 | 0.24 |
| 256 | **21.1** | 5.1 | 0.24 | **20.2** | 4.9 | 0.25 | **16.1** | 4.2 | 0.26 |

Table 9: Detailed rank dependence results for DoRA on the Pile of Law dataset; metrics computed as in Table 5. **Emphasis** marks the best overall performance.

# H EXTENDED PEFT PERFORMANCE

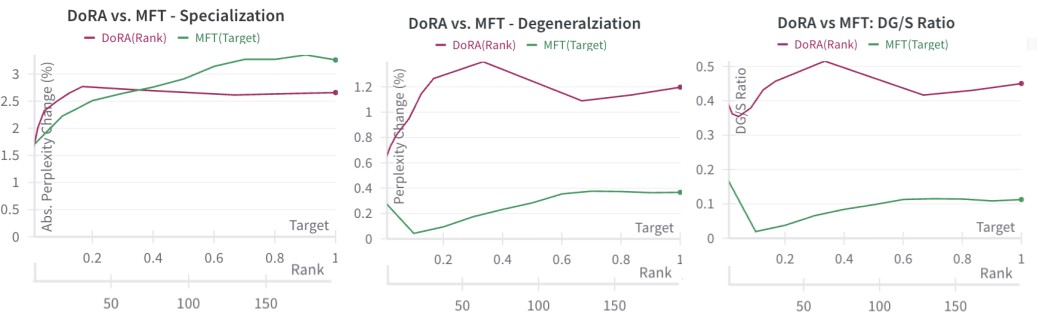

Figure 7: DoRA with increasing rank set against MFT with increasing target on the OpenWebMath dataset. *Left.* MFT outperforms DoRA in specialization for sufficiently large targets. *Middle.* MFT consistently exhibits lower degeneralization than DoRA across all targets and ranks. *Right.* As a result, MFT exhibits 50-90% lower $DG/S$ ratios than DoRA.

| Model | Method | Dataset | | | | | | | | |
|---|---|---|---|---|---|---|---|---|---|---|
| | | PubMed | | | Pile of Law | | | OpenWebMath | | |
| | | $S\uparrow$ | $DG\downarrow$ | ratio $\downarrow$ | $S\uparrow$ | $DG\downarrow$ | ratio $\downarrow$ | $S\uparrow$ | $DG\downarrow$ | ratio $\downarrow$ |
| Gemma 2B | LoRA | **4.2** | 1.2 | 0.28 | **4.9** | 1.4 | 0.29 | **1.3** | 0.6 | 0.44 |
| | LoRA+MFT | 2.6 | **0.2** | **0.09** | 3.2 | **0.3** | **0.10** | 0.7 | **0.1** | **0.15** |
| | DoRA | **4.2** | 1.2 | 0.28 | **4.9** | 1.4 | 0.29 | **1.3** | 0.6 | 0.46 |
| | DoRA+MFT | 2.6 | **0.2** | **0.09** | 3.2 | **0.3** | **0.11** | 0.7 | **0.1** | **0.15** |
| | IA3 | **1.0** | 0.1 | 0.06 | **1.4** | 0.1 | 0.04 | **0.3** | 0.0 | 0.04 |
| | IA3+MFT | 0.9 | **0.0** | **0.04** | 1.0 | **0.0** | **0.04** | **0.3** | **0.0** | **0.03** |
| Minitron 4B | LoRA | **7.6** | 2.3 | 0.30 | **5.8** | 2.1 | 0.36 | **0.6** | 0.3 | 0.46 |
| | LoRA+MFT | 5.5 | **0.6** | **0.11** | 4.4 | **0.7** | **0.15** | 0.3 | **0.1** | **0.40** |
| | DoRA | **7.7** | 2.3 | 0.30 | **5.8** | 2.1 | 0.35 | **0.6** | 0.2 | 0.43 |
| | DoRA+MFT | 5.5 | **0.6** | **0.11** | 4.4 | **0.6** | **0.15** | 0.3 | **0.1** | **0.24** |
| | IA3 | **1.0** | 0.0 | 0.04 | **1.2** | 0.1 | 0.06 | **0.0** | 0.0 | - |
| | IA3+MFT | 0.7 | **0.0** | 0.06 | 0.9 | **0.1** | 0.07 | **0.0** | **0.0** | - |
| LLaMA 2 7B | LoRA | **7.7** | 0.8 | 0.10 | **6.1** | 0.7 | 0.12 | **2.5** | 0.6 | 0.22 |
| | LoRA+MFT | 5.5 | **0.2** | **0.03** | 4.3 | **0.1** | **0.03** | 1.7 | **0.1** | **0.08** |
| | DoRA | **7.8** | 0.8 | 0.10 | **6.1** | 0.7 | 0.12 | **2.6** | 0.6 | 0.24 |
| | DoRA+MFT | 5.5 | **0.2** | **0.03** | 4.3 | **0.1** | **0.02** | 1.7 | **0.1** | **0.08** |
| | IA3 | **1.2** | 0.0 | 0.02 | **1.8** | 0.0 | 0.02 | **0.6** | 0.0 | 0.05 |
| | IA3+MFT | **1.2** | **0.0** | **0.02** | 1.5 | **0.0** | **0.02** | 0.3 | **0.0** | **0.04** |

Table 10: Evaluation of MFT against FT when used in conjunction with LoRA/DoRA/IA3 across larger language models and several specialized-domain datasets; metrics computed as in Section 3.1. $S, DG$ are reported in percentages, $DG/S$ is reported as a fraction. **Emphasis** marks the better value for each pair of experiments.

