# OpenReview forum: "Minifinetuning: Low-Data Generation Domain Adaptation through Corrective Self-Distillation"
_ICLR.cc/2025/Conference — Submitted to ICLR 2025_

### Official Review · Reviewer_m7QG · 2024-10-29

**Soundness:** 2
**Presentation:** 2
**Contribution:** 3
**Rating:** 5
**Confidence:** 4

**Summary:**

This paper presents a new method, minifinetuning, that supports autoregressively finetuning LLMs on specialized domains while attempting to mitigate the amount of catastrophic forgetting of the pretraining corpus.

Minifinetuning makes use of a teacher-student setup, where the student is fine-tuned on a "corrected" token distribution, which is a hybrid of the new domain's token distribution the token probabilities output by the teacher. In this way, each token's distribution remains close to the original pretraining distribution (according to the teacher), with a minimal modification to boost the probability of the actual target token in the specialized domain.

The authors evaluate multiple models (GPT-Neo, Phi, Gemma, and Llama 2) on three specialized subdomains (medical, legal, and math), comparing against both vanilla finetuning and parameter-efficient finetuning baselines. Replaying samples from the pretraining data tends to outperform all methods considered, but the authors argue that access to pretraining samples amounts to an "unfair advantage." Their results suggest that minifinetuning allows for appropriate specialization to the new domain, while mitigating (to a certain extent) performance degradations on the pretraining data. This is particularly the case when the pretraining with limited data.

**Strengths:**

- Minifinetuning tackles the important problem of catastrophic forgetting of the pretraining data, though the authors could more clearly motivate why one wants to preserve the finetuning distribution after specialized finetuning.
- The proposed approach is simple, and the benchmarks and baselines considered are sufficiently comprehensive to evaluate its performance.
- The authors provide a helpful ablation study which shows the effect of adjusting only those token distributions that result in the incorrect token on the specialized domain, providing support for the author's implemention of minifinetuning (where all distributions are modified, whether resulting in the correct or incorrect token on the specialized domain).

**Weaknesses:**

- There is insufficient detail on how hyperparameters were selected for both minifinetuning and for the baselines. Why was Tau set to 0.25? How would the authors suggest setting this in practice; based on the specialization, degeneralization, ratio, or something else?
- Similarly, for the LoRA and DoRA baselines, how did the authors decide to fix the rank at 8? To make robust comparisons between minifinetuning and e.g. LoRA, it is important to understand whether this is a best-vs-best comparison. If Tau was tuned, but rank was not, then this isn't a fair comparison.
- As mentioned in Strengths, the paper should more clearly motivate why a practitioner might want to conduct specialization without degrading pretraining performance.
- The paper needs some significant attention to improve its clarity, and there are numerous errors scattered throughout. For example, the phrase "construction of a per-token individualized corrected distribution" is very hard to parse, as are the italicized subheadings under section 4, e.g., "distillation and corrective distillation on incorrect tokens". I had to read several sections multiple times.
- **Minor**: Is "degeneralization" a new term coined by the authors? Perhaps "performance decrease" or "catastrophic forgetting" might more clearly convey the underlying idea and tie in with prior work.
- **Minor**: Subfigures often have axes on different scales. It would be helpful to use a shared scale to facilitate easy visual comparison, or to explicitly call out a 10x decrease in y axis range (e.g. fig. 1).
- **Minor**: The tables are rather large, and it is difficult to draw clear conclusions, particularly given the presence of Replay (e.g. Table 2). Perhaps the authors could add a short takeaway message to the caption of each table, highlighting what one should be looking for.

**Questions:**

1. The underlying idea significantly resembles how RLHF-based post-training is conducted with PPO, wherein a KL divergence term controls distribution shift between the preference-tuned model and the original base model. Did the authors consider interpolating between the full teacher distribution and the full student distribution, rather than just w.r.t. the target token?
2. Do the authors have any intuition as to why MFT appears to improve specialization, rather than just attenunate degeneralization (e.g. fig 3, left top and bottom, 1M and 2M tokens)?

---

> ### Author Response · Authors · 2024-11-19
> **Author Response**
>
> Dear Reviewer m7QG,
>
> Thank you for your thorough review. In response to the points raised:
>
> > There is insufficient detail on how hyperparameters were selected for both minifinetuning and for the baselines. Why was Tau set to 0.25? How would the authors suggest setting this in practice; based on the specialization, degeneralization, ratio, or something else?
>
> $\tau$ is not a hyperparameter per se, it is a design decision, as noted on L158 and L316-317. The user is free to choose $\tau$ as needed, in accordance with their appetite for specialization and tolerance for degeneralization.
>
> > Similarly, for the LoRA and DoRA baselines, how did the authors decide to fix the rank at 8? To make robust comparisons between minifinetuning and e.g. LoRA, it is important to understand whether this is a best-vs-best comparison. If Tau was tuned, but rank was not, then this isn't a fair comparison.
>
> $\tau$ was not tuned, because there does not exist an objectively better value of $\tau$ in any sense. One can either opt in for higher values of $\tau$ (meaning better specialization but higher degeneralization) or lower values (meaning lower specialization but also better preservation of general performance).
>
> It would appear that you are suggesting that the values of $\tau$ and LoRA/DoRA ranks have been chosen to deliberately show MFT in a better light. Please revisit Figure 2 and Appendices E-H to convince yourself that this is indeed *not the case*. In particular, Figure 2 visually demonstrates that the DoRA degeneralization-specialization dynamic is at all times well-separated from that of MFT.
>
> > As mentioned in Strengths, the paper should more clearly motivate why a practitioner might want to conduct specialization without degrading pretraining performance.
>
> We concede that this motivation is currently only hinted on at lines 31-36. We will revisit the manuscript to better point out the motivation.
>
> > The paper needs some significant attention to improve its clarity, and there are numerous errors scattered throughout. For example, the phrase "construction of a per-token individualized corrected distribution" is very hard to parse, as are the italicized subheadings under section 4, e.g., "distillation and corrective distillation on incorrect tokens". I had to read several sections multiple times.
>
> Thank you for bringing this to our attention. Indeed, the first example is missing a comma between coordinate adjectives. We have further edited the Ablation Study subsections of Section 4.
>
> > Minor: Is "degeneralization" a new term coined by the authors? Perhaps "performance decrease" or "catastrophic forgetting" might more clearly convey the underlying idea and tie in with prior work.
>
> The term has appeared in related work, albeit only colloquially. We have chosen to adopt it to over "performance decrease" due to its terseness. We also note that unlike most current catastrophic forgetting metrics, our metrics are perplexity- rather than accuracy-based. Furthermore, low-resource training and finetuning is a separate subfield of LLM post-training with only distant links to continual learning, where the term "catastrophic fogetting" is ubiquitous. (Please note the lack of method comparison to continual learning methods in the directly relevant related work, e.g.  Li et al., 2020; Chronopoulou et al., 2021; Diao et al., 2023; Huang et al., 2023; Yang et al., 2024). Hence our choice of degeneralization/specialization over catastrophic forgetting/"learning".
>
> > Minor: Subfigures often have axes on different scales. It would be helpful to use a shared scale to facilitate easy visual comparison, or to explicitly call out a 10x decrease in y axis range (e.g. fig. 1).
>
> We experimented with both sharing the axis range across all figures and merging the figures into one while employing logarithmic scaling. The former consistently squashed the curves of the less-performing figures to the point where they became indistinguishable, whereas the latter succeeded in capturing the relative relationship between the two sets of curves but became an impediment to the analysis of individual curves within one set. We understand the natural longing for increased consistency, but it would only come at the cost of the figures becoming less informative with regard to the claims made in the main tax, which we do not wish to compromise. We are open to more discussion on the topic, however.
>
> > Minor: The tables are rather large, and it is difficult to draw clear conclusions, particularly given the presence of Replay (e.g. Table 2). Perhaps the authors could add a short takeaway message to the caption of each table, highlighting what one should be looking for.
>
> We included such takeaways in bold for each table; for Table 2, this were the bold paragraph starts across lines 255-307. Please let us know if any of the takeaways need particular attention.

---

> ### Author Response · Authors · 2024-11-19
> **Author Response**
>
> > The underlying idea significantly resembles how RLHF-based post-training is conducted with PPO, wherein a KL divergence term controls distribution shift between the preference-tuned model and the original base model. Did the authors consider interpolating between the full teacher distribution and the full student distribution, rather than just w.r.t. the target token?
>
> The answer is yes and no.  On the "no" side, one cannot engage the student distribution alone, because at the beginning of the training the student and the teacher are identical, so there would never be any progress. We did however try to train the student model by traditional finetuning while also punishing deviations from the teacher by KL divergence. This result consistently resulted in suboptimal solutions, in which the component of the loss that was due to ground truth cross entropy became effectively the same as the component of the loss due to KL divergence, and the training stalled. We then felt that there were no guarantees on the nature of the student distributions after training, and decided to impose more control by explicitly specifying new "corrected" target distributions.
>
> > Do the authors have any intuition as to why MFT appears to improve specialization, rather than just attenunate degeneralization (e.g. fig 3, left top and bottom, 1M and 2M tokens)?
>
> There are two values to observe in Figure 3 left top and bottom for 1M and 2M series. One is the peak specialization, which is actually still better for FT than for MFT for both series. The other one, which we think is the one you're referring to is the tail specialization, i.e. the behavior that comes in after the model begins to overfit. In this case, the intuition is the same as for MFT as whole; MFT prevents destructive generalization because of the variable nature of the target offsetting by $\tau$. Since the training targets (in this case the corrected distributions) are always corrected to only allow the ground truth to surpass the current most likely token by $\tau$, they still continue to convey a considerable amount of information about the original teacher (i.e. untrained) distribution, thus keeping overfitting at bay.
>
> In other words, the apparent outperfomance of FT by MFT in terms of the tail specialization when finetuning data is very scarce is a consequence of MFT anchoring the training by the teacher distribution at all times, i.e. an intended consequence of the MFT's design decisions.
>
> We stand available for further discussion.

---

> > ### Author Response · Authors · 2024-11-22
> > **Further Author Response**
> >
> > Dear Reviewer m7QG,
> >
> > Please see the above response for additional clarification and explanations where solicited. We continue to be available for discussion, please let us know if there are other points that require our attention.

---

> > > ### Comment · Reviewer_m7QG · 2024-11-25
> > >
> > > Thank you for your response to my comments and questions.
> > >
> > > I am unconvinced the the argument that tau is a "design decision", rather than a hyperparameter. Ultimately, a practitioner will need to set this, and it appears (from figure 2) to influence the specialization/degeneralization trade-off.
> > >
> > > In light of your response, I would strongly advocate that you use standard terminology, "catastrophic forgetting", to refer to degeneralization. Catastrophic forgetting refers to performance degradation regardless of metric or task, and there's really no need to use something else here.
> > >
> > > Finally, thank you for highlighting your summaries of table takeaways, e.g. lines 255-311. I would suggest that the authors incorporate a succint summary into the table captions themselves, such that it is immediately clear what to look at without referring to the main text. This is also the case for each of the tables in the appendix.

---

> ### Author Response · Authors · 2024-11-27
> **Further Author Response**
>
> Dear Reviewer m7QG,
>
> > I am unconvinced the the argument that tau is a "design decision", rather than a hyperparameter. Ultimately, a practitioner will need to set this, and it appears (from figure 2) to influence the specialization/degeneralization trade-off.
>
> As we outlined in our response above and in great detail in Section 2 of the paper, $\tau$ is the choice of the method's user, just like the choice of the transformer hidden dimension, the number of attention heads in the MHA mechanism, or the choice of LoRA rank in LoRA finetuning. Its value has consequences, but it is not a tunable hyperparameter because there is no implicit notion of optimality to tune it for. Both lower and higher values of $\tau$ have their uses, and it boils down to what the user of the method is trying to achieve. $\tau=1$ is equivalent to classical finetuning, $\tau=0$ is the minimal correction necessary to bring the likelihood of the ground truth finetuning token on par with the most likely prediction of the unfinetuned model. As such, please note that the above response was not an argument, and that our explanation of the $\tau$'s role is a statement of a fact supported by $\tau$'s formal description Section 2 rather than an attempt to convince the reader.
>
> > In light of your response, I would strongly advocate that you use standard terminology, "catastrophic forgetting", to refer to degeneralization. Catastrophic forgetting refers to performance degradation regardless of metric or task, and there's really no need to use something else here.
>
> It would appear that our response above went partially unnoticed. To develop an understanding of the subfield, we strongly encourage revisiting our overview of related work on low-resource finetuning in Section 6. Please note in particular the lack of method comparison to continual learning methods and scarcity of references to "catastrophic forgetting" in the directly relevant work, e.g. Li et al., 2020; Chronopoulou et al., 2021; Diao et al., 2023; Huang et al., 2023; Yang et al., 2024.
>
> Consistently with the practice in low-resource finetuning and LLM post-training more broadly, and abiding by the precedent set by our colleagues in the field, we will not be renaming our perplexity-based measure of degeneralization to "catastrophic forgetting". Unlike the simpler accuracy-based measures of forgetting proposed in CL, an increase in perplexity does not necessarily imply the occurrence of forgetting in LLM finetuning, further justifying the literature's reluctance to refer to the term.
>
> > Finally, thank you for highlighting your summaries of table takeaways, e.g. lines 255-311. I would suggest that the authors incorporate a succint summary into the table captions themselves, such that it is immediately clear what to look at without referring to the main text. This is also the case for each of the tables in the appendix.
>
> Thank you for raising this point. We edited the captions for the rebuttal version of the manuscript.

---

### Official Review · Reviewer_ZYc9 · 2024-11-03

**Soundness:** 3
**Presentation:** 3
**Contribution:** 3
**Rating:** 6
**Confidence:** 4

**Summary:**

This paper proposes MINI-FINETUNING, a fine-tuning technique that could balance specification and generalization without relying on pre-training data. The proposed technique uses the output probability of un-finetuned model as a teacher distribution to regularize the fine-tuning objective.

**Strengths:**

1. This presentation is clear.
2. The method is straightforward and intuitive.

**Weaknesses:**

1. Although most parts of the method can be justified, I find that one particular design is a bit heuristic and unexplained. In particular, the method essentially involves a weighted combination of the one-hot distribution (for fine-tuning) and the original model's output distribution (for avoiding degeneralization). However, instead of a constant $\alpha$, the authors adopted a variable $\alpha$, such that the correct answer for the fine-tuning always dominates the original model's best answer by a fixed margin, $\tau$ (when the two answers are different). The motivation behind this is not well-explained. To me, this may induce undesirable behaviors. Consider two cases where the original model's answer differs from the fine-tuning label. In the first case, the original model's prediction is very certain, thus the distribution is very sharp; in the second case, the original model's prediction is very uncertain, thus the distribution is very flat. My intuition is that we would want to preserve the original model's knowledge more in the first case, because the knowledge is clear and very concentrated; and we would want to adapt to the fine-tuning objective more in the second case, because the original model doesn't hold a strong belief. However, the proposed method would do the opposite -- it would down-weight the original model's prediction more for the first case, because it needs to suppress the sharp distribution more to achieve a sufficient margin. This may lead to the model forgetting the key, definitive knowledge in the original model, and only retaining the superficial, ambiguous knowledge. This design is also at odds with the Replay method, to which the authors make an analogy to justify the proposed method. The Replay method can be regarded as having a fixed weight for all tokens. Could the authors better explain why it is necessary to deviate from Replay's intuition and have a variable weight, or at least provide some empirical evidence that the proposed weighting is better?

2. The authors seemed to experiment primarily with parameter-efficient LLMs. However, the specification-generalization trade-off is at greater stake and more challenging for regular state-of-the-art LLMs, such as Llama 3, 3.1 or 3.2, Mistral, Mixtral, etc. The authors did experiment with larger models in the appendix, but oddly with Llama 2 7B only, which is already outdated. Is there a reason not to test on more recent, state-of-the-art open-sourced models?

3. The authors seem to miss the rich body of works on catastrophic forgetting, which studies how to learn new knowledge without forgetting the old one, such as [1]. It would be very helpful if the authors could discuss those works in the related work section, and discuss the contribution of the proposed method in the context of catastrophic forgetting. Otherwise, adding self-distillation to prevent catastrophic forgetting does not constitute a very novel contribution.

4. (Minor) Table 1 is neither referenced nor explained in the paper. What do the stars mean? How were they rated?


[1] Kirkpatrick, James, et al. "Overcoming catastrophic forgetting in neural networks." Proceedings of the national academy of sciences 114.13 (2017): 3521-3526.

**Questions:**

Please see the weaknesses section.

---

> ### Author Response · Authors · 2024-11-19
> **Author Response (Part 1)**
>
> Dear Reviewier ZYc9,
>
> Thank you for your review. In response to the points raised:
>
> > However, instead of a constant $\alpha$, the authors adopted a variable $\tau$, such that the correct answer for the fine-tuning always dominates the original model's best answer by a fixed margin, (when the two answers are different). The motivation behind this is not well-explained.
>
> Please revisit Section 2, especially lines 153 to 164. The corrections are done on per-token basis, meaning that aiming to “improve by target $\tau$” results in variable values or $\alpha$ and $\beta$ depending on the situation. $\alpha$ is not a constant as claimed by the review, and explanation of the motivation behind doing so is outlined on lines 75-82 and 95-100. That this is necessary is then empirically proven in on lines 391-449. We are available for clarification of concrete detail if needed.
>
> > Consider two cases where the original model's answer differs from the fine-tuning label. In the first case, the original model's prediction is very certain, thus the distribution is very sharp; in the second case, the original model's prediction is very uncertain, thus the distribution is very flat. My intuition is that we would want to preserve the original model's knowledge more in the first case, because the knowledge is clear and very concentrated; and we would want to adapt to the fine-tuning objective more in the second case, because the original model doesn't hold a strong belief. However, the proposed method would do the opposite -- it would down-weight the original model's prediction more for the first case, because it needs to suppress the sharp distribution more to achieve a sufficient margin. This may lead to the model forgetting the key, definitive knowledge in the original model, and only retaining the superficial, ambiguous knowledge.
>
> This is incorrect. Note that knowledge of the original model is represented by its whole distribution, and that in both cases you mention (concentrated and dispersed probabilities), MFT reduces the mass of the original distribution uniformly and by the minimum factor necessary to allow the new, desired ground truth to be more likely by an offset $\tau$. As detailed on lines 153-160, this is the minimal adjustment *necessary* in order for the finetuning objective (namely for the ground truth token to become most likely) to be accomplished. As such, our method outperforms traditional finetuning (which completely erases knowledge in both cases). Note also that entropy of the token distribution does not, broadly speaking, represent the strength of model belief in language modeling (think of ambiguous phrasing). However, if you know of a work that experiments with the method you propose, we would welcome a reference for its inclusion in Section 6.
>
> > This design is also at odds with the Replay method, to which the authors make an analogy to justify the proposed method.
>
> Please clarify how this design is at odds with the Replay method. We show in the paper this method is complementary -- Replay can be composed MFT.
>
> > Is there a reason not to test on more recent, state-of-the-art open-sourced models?
>
> We do indeed test on recent, state-of-the-art open-sourced models. The heaviest brunt of the experimentation is borne by Apple OpenELM family (April 2024) which exhibits best-of-size performance on many tasks, and the experimentation is further supported by the evaluation on the most recent Nvidia Minitron 4B (August 2024). To hold experimentation on Llama 2 7B as an example of us not experimenting on “more challenging regular state-of-the-art LLMs” is not to faithfully report on our experimentation.
>
> Please also note that Llama 3.2 was released less than a week before the ICLR'25 deadline.
>
> > The authors seem to miss the rich body of works on catastrophic forgetting, which studies how to learn new knowledge without forgetting the old one, such as [1]. It would be very helpful if the authors could discuss those works in the related work section, and discuss the contribution of the proposed method in the context of catastrophic forgetting. Otherwise, adding self-distillation to prevent catastrophic forgetting does not constitute a very novel contribution.
>
> The reviewer seems to miss the rich discussion of related work in Section 6, in particular the nature and objectives of directly relevant works such as Li et al., 2020; Chronopoulou et al., 2021; Diao et al., 2023; Huang et al., 2023; Yang et al., 2024. Our work is focused on low-data finetuning of language models, and we do not aim to nor claim to propose a new continual learning method. We encourage the reviewer to inspect these references and note the dearth of comparisons to continual learning methods.

---

> > ### Comment · Reviewer_ZYc9 · 2024-11-19
> > **Further comment**
> >
> > Dear authors,
> >
> > I feel that there is a misunderstanding regarding the weaknesses 1. I would like to clarify my points, so that authors get more opportunities to respond.
> >
> > 1. Let's assume a vocabulary of 3, for simplicity. Let's say that at one token, the original model's output probability over the three tokens is [0.9, 0.05, 0.05], which happens when the original model is very confident about its answer. Let's say at the second token position, the output probability over the three tokens is [0.5, 0.2, 0.3], which happens when the original model is not quite confident about its prediction of the next token. Now, let's assume that in both cases, the finetuning data set's answers are different from the model's original prediction. Then, the $\alpha$ for the first case is larger than the $\alpha$ for the second case, because $1-\alpha$ needs to be larger in the first case to suppress the 0.9. However, this may not be the optimal choice because in the first case, the [0.9, 0.05, 0.05] distribution signals that the original model holds a strong belief of its prediction, so even though the finetuning's answer is different, we may not want to downweight the original distribution by that much (otherwise this may hurt the original model's performance). In short, always suppressing the original model's top response to gain a fixed margin may encourage the model to drop its confident beliefs. Have the authors try a constant $\alpha$ instead?
> >
> > 2. I acknowledge the authors' experiments on state-of-the-art parameter-efficient LLMs, such as Minitron 4B. My point was experiments on larger-sized, more recent models are still necessary because the challenge of retaining the original model's knowledge may be greater for larger models. It would be helpful if the authors could report the results on Llama 3 or 3.1 7B, mistral 7B.

---

> ### Author Response · Authors · 2024-11-19
> **Author Response (Part 2)**
>
> > (Minor) Table 1 is neither referenced nor explained in the paper. What do the stars mean? How were they rated?
>
> Please revisit lines 91 to 94 find the explanation of Table 1. Stars denote levels of quality, as is common in hotel, restaurant, or general consumer reviews. The star ratings work as follows: the best method in class receives the maximum number of starts (in this case 3), while the least performant method receives 0 stars. The better a method is relative to others, the more stars it receives.
>
> We stand available for further discussion.

---

> > ### Comment · Reviewer_ZYc9 · 2024-11-19
> > **Further comments**
> >
> > Again, the concern is not fully addressed, although this is a minor one. Table 1 should have been referenced in the main text. Do the authors have plans to refer to this table somewhere in the main text? The captions (lines 91 to 94) do not fully explain how the stars were rated. Do the authors plan to refine the captions? Also, the author's explanation in the rebuttal is still vague. 'The better a method is relative to others, the more stars it receives'. How did the authors *quantify* the stars? Is it a linear or nonlinear interpolation in terms of the performance? What metrics were used to compute the interpolation?

---

> > > ### Author Response · Authors · 2024-11-21
> > > **Further Author Response (Part 3)**
> > >
> > > > Again, the concern is not fully addressed, although this is a minor one. Table 1 should have been referenced in the main text. Do the authors have plans to refer to this table somewhere in the main text?
> > >
> > > The purpose of Table 1 is to give a high-level overview of the relative strengths and weaknesses of the methods considered in further down this work. As such, it is of qualitative rather than quantitative nature by design. The table does not introduce any new results that are not otherwise covered in later tables and figures.
> > >
> > > > The captions (lines 91 to 94) do not fully explain how the stars were rated. Do the authors plan to refine the captions? Also, the author's explanation in the rebuttal is still vague. 'The better a method is relative to others, the more stars it receives'. How did the authors quantify the stars?
> > >
> > > With regard to the stars: as is common in recent work, the stars are assigned based on the rounded-up value of the that they are to describe (in this case the specialization $S$ and degeneralization $DG$. More formally, the stars the per-point values of the injection from the reals reported in Table 2 for each metric to the discrete set $\{0, 0.5, 1, 1.5, 2, 2.5, 3\}$  such that the weak order on $\mathbb{R}$ is preserved. Where needed to break ties and guarantee uniqueness, the pairwise euclidean distance between the source reals is used to determine the pairwise relative distances in the injection's image.
> > >
> > > > What metrics were used to compute the interpolation?
> > >
> > > Please revisit the first column of Table 1, marked as "Property", to see the names of the metrics considered.
> > >
> > > We agree that this is a minor thing and would be willing to replace the stars in Table 1 with ranges for the metrics considered copies from Table 2. Would this alleviate your concern?
> > >
> > > As before, we stand available for further discussion.

---

> > > > ### Comment · Reviewer_ZYc9 · 2024-11-22
> > > > **Thanks for the response**
> > > >
> > > > I would like to thank the reviewers for the detailed response. The discussion and new results resolved many concerns I had. I have adjusted my scores accordingly. Regarding Table 1, I would encourage the authors to either elaborate on how the stars are rated, as discussed in the response, or replace the stars with ranges for the metrics.

---

> ### Author Response · Authors · 2024-11-21
> **Further Author Response (Part 1)**
>
> Dear Reviewer
>
> Thank you for your clarification, we now better understand the source of the confusion.
>
> > Let's say that at one token, the original model's output probability over the three tokens is [0.9, 0.05, 0.05], which happens when the original model is very confident about its answer.
>
> Although not essential here; as we noted above, the entropy of the probability distribution is not a reliable measure of model's confidence. A model can provide a high-entropy output token distribution even when it very certain about its generation, especially in cases when a phrase is ambiguous or the order of the outputs does not matter as is commonly seen in code generation.
>
> >  Let's say at the second token position, the output probability over the three tokens is [0.5, 0.2, 0.3], which happens when the original model is not quite confident about its prediction of the next token. Now, let's assume that in both cases, the finetuning data set's answers are different from the model's original prediction. Then, the  $\alpha$ for the first case is larger than the $\alpha$ for the second case, because needs to be larger in the first case to suppress the 0.9. However, this may not be the optimal choice because in the first case, the [0.9, 0.05, 0.05] distribution signals that the original model holds a strong belief of its prediction, so even though the finetuning's answer is different, we may not want to downweight the original distribution by that much (otherwise this may hurt the original model's performance).
>
> The issue with this intuition is that "suppressing" the original distribution by a variable factor $\alpha$ is not a question of optimality, but necessity. As detailed in Section 2 (lines 153-172), $\alpha$ is always chosen to be such that the finetuning ground truth target tokens is more likely by $\tau$. If $\alpha$ were a fixed constant, it could be that a zero-loss model would still not favour the ground truth token over its original (teacher) prediction, so the finetuning target of tuning the model on the new data would not be accomplished.
>
> We reiterate for clarity that this is a finetuning work. The goal of the MFT is to tune the model on a new dataset, or more precisely, to train with the objective that its generation predictions match the ground truth on the data as much as possible (with the additional design requirement for as little degeneralization as possible). No notion of optimality applies here as there is no other way to choose $\alpha$ given the objective to train the model such that its predictions match that of the ground truth. You can argue that one can train toward a fixed mixture by a constant $\alpha$, but this would be training the model toward a different objective and not the finetuning objective. As noted in the previous response, while it was our initial hope that such a constant would exist, we found the level of success of various constant to be highly dependent on the few samples being used in our low-data setting and therefore unsuitable for use as a general method, as such $\alpha$ would have to be found by an exhaustive search for every finetuning instance. The use of the target offset $\tau$ was the next natural and necessary step to create a general method whose parametrization does not suffer from heavy dependency on the finetuning data used.

---

> ### Author Response · Authors · 2024-11-21
> **Further Author Response (Part 2)**
>
> > I acknowledge the authors' experiments on state-of-the-art parameter-efficient LLMs, such as Minitron 4B. My point was experiments on larger-sized, more recent models are still necessary because the challenge of retaining the original model's knowledge may be greater for larger models. It would be helpful if the authors could report the results on Llama 3 or 3.1 7B, mistral 7B.
>
> Please note that two of the models you mention, namely Llama 3 7B and Llama 3.1 7B, do not exist. However, we heed your request, and have therefore run additional experiments with the latest and closest available variants, in particular Llama 3.1 8B, Llama 3.2 1B, and Llama 3.2 3B. Please find the results below.
>
> |              |        | PbMd |     |       | PoL |     |       | OWM |     |       |
> | ------------ | ------ | ------ | --- | ----- | ----------- | --- | ----- | ----------- | --- | ----- |
> | Model        | Method | S      | DG  | Ratio | S           | DG  | Ratio | S           | DG  | Ratio |
> | Llama 3.1 8B | FT     | 7.0 | 1.4 | 0.20  | 6.4         | 1.5 | 0.23  | 2.6         | 0.9 | 0.35  |
> |              | LoRA   | 6.8    | 1.8 | 0.26  | 5.2         | 1.2 | 0.23  | 2.8         | 0.7 | 0.25  |
> |              | DoRA   | 6.9    | 1.6 | 0.23  | 5.0         | 1.1 | 0.22  | 2.8         | 0.9 | 0.26  |
> |              | IA3    | 1.2    | 0.1 | 0.08  | 1.5         | 0.0 | 0.01  | 0.6         | 0.0 | 0.08  |
> |              | MFT    | 6.6    | 0.9 | 0.14  | 5.1         | 0.4 | 0.08  | 2.3         | 0.4 | 0.17  |
> |              | Replay | 6.4    | 0.1 | 0.02  | 5.4         | 0.2 | 0.04  | 2.3         | 0.2 | 0.09  |
> | Llama 3.2 1B | FT     | 11.2   | 3.5 | 0.31  | 12          | 5.2 | 0.43  | 2.1         | 1.0 | 0.48  |
> |              | LoRA   | 10.8   | 3.8 | 0.35  | 9.9         | 5.0 | 0.51  | 2.0           | 0.9 | 0.45  |
> |              | DoRA   | 10.8   | 3.2 | 0.30  | 9.9         | 5.0 | 0.51  | 1.9         | 0.8 | 0.42  |
> |              | IA3    | 2.3    | 0.4 | 0.17  | 1.7         | 0.1 | 0.06  | 0.1         | 0.0 | 0.04  |
> |              | MFT    | 10.5   | 1.7 | 0.16  | 9.7         | 1.5 | 0.15  | 1.6         | 0.3 | 0.19  |
> |              | Replay | 7.5    | 1.5 | 0.20  | 11.3        | 1.0 | 0.09  | 2.1         | 0.0 | 0.02  |
> | Llama 3.2 3B | FT     | 9.1    | 1.6 | 0.18  | 8.3         | 2.8 | 0.34  | 4.0         | 1.9 | 0.48  |
> |              | LoRA   | 8.2    | 1.6 | 0.20  | 7.8         | 2.6 | 0.33  | 3.3         | 1.7 | 0.52  |
> |              | DoRA   | 8.2    | 1.5 | 0.18  | 7.8         | 2.6 | 0.33  | 3.3         | 1.7 | 0.52  |
> |              | IA3    | 2.1    | 0.2 | 0.10  | 1.7         | 0.1 | 0.06  | 0.3         | 0.1 | 0.33  |
> |              | MFT    | 7.6    | 0.9 | 0.12  | 7.5         | 0.8 | 0.11  | 3.1         | 0.4 | 0.13  |
> |              | Replay | 6.8    | 0.4 | 0.06  | 7.9         | 0.6 | 0.08  | 3.3         | 0.1 | 0.03  |
>
> Upon inspection, we found these results broadly consistent with the findings of Section 3.2.

---

### Official Review · Reviewer_G3zW · 2024-11-04

**Soundness:** 2
**Presentation:** 2
**Contribution:** 2
**Rating:** 5
**Confidence:** 4

**Summary:**

This paper develops a novel technique for intelligent fine-tuning while minimizing forgetting of model representations learned on the pre-training dataset. The main proposed technique titled minifinetuning (MFT) employs a teacher, student based training paradigm where the (frozen) teacher model represents the pre-trained model while the (learnable) student model is initialized with the teacher weights but updated intelligently with the proposed technique to learn representations of the fine-tuning data. The crux of the finetuning technique is based on the idea of "moving" density mass uniformly from the prediction distribution P^T of a teacher model to the required target word as described by the ground truth (one-hot encoded) label on the fine-tuning dataset. The student model is updated via. crossentropy loss to mimic this corrected version of P^T to better align with the fine-tuning dataset. Instead of directly fine-tuning fully the student model via. cross-entropy loss on one-hot encoded labels, the intuition of this approach is that the degree to which P^T is modified can be controlled by user-specified parameters thereby striking a better balance between forgetting representations from the pre-training dataset and generalizing well to the fine-tuning dataset. Although the method is interesting, some crucial comparisons with existing methods that minimize catastrophic forgetting are missing as are evaluation with existing evaluation metrics popularly used in the fine-tuning, continual learning literature. Further, the related work needs to be more comprehensive and the visualization quality needs to be improved.

**Strengths:**

1. The paper introduces interesting results with a new, simple methodology for fine-tuning with minimal perturbation of the existing distribution learned by language models. The proposed method is intuitive and results (albeit incomplete) seem to indicate promising balance between retaining pre-trained representations while learning appropriate representations on the fine-tuning data.

2. The core idea of the soft KD by uniformly moving density mass to a specific location of interest to better match the target probability distribution is an interesting formulation and could lead to more sophisticated improvements using optimal transport based techniques in the future.

**Weaknesses:**

1. The coverage of techniques that alleviate catastrophic forgetting needs to be more extensive and the paper largely ignores a large set of techniques based on alleviating catastrophic forgetting and also metrics to measure forgetting (e.g., see Sec. 2.3 in `[2]`) and the stability-plasticity tradeoff (sec. 3.1 in `[2]`).

2. The paper has a weak baseline comparison and misses comparing with state-of-the-art papers in regularization based continual learning to mitigate catastrophic forgetting (e.g., `[1]`). Such continual learning style papers are applicable as the authors make it a point many times in the paper (e.g., last row of Table 1) to highlight that the major advantage of their proposed MFT method relative to `Replay-based` methods is that MFT doesn't rely on access to training data thereby making the fine-tuning scenario similar to a continual learning style adaptation. In such a context, specific, approaches (e.g., `[1]`) exist that only retain metrics of weight-level `importance' on the pre-training dataset. This importance score is employed to intelligently fine-tune the model to minimize forgetting.

3. Paper presentation and specifically visualizations should be significantly improved. Multiple qualitative comparisons are drawn between plots where the y-axis has values on different scales (e.g., Fig. 1b,c; Fig. 3 FT:Ratio, MFT:Ratio; FT:Degeneralization, MFT:Degeneralization etc.). Either the scales should be unified (e.g., via. log scaling the axis) or such figures can be re-designed as tables for easier comparison.

## References:

1. James Kirkpatrick, Razvan Pascanu, Neil Rabinowitz, Joel Veness, Guillaume Desjardins, Andrei A Rusu, Kieran Milan, John
Quan, Tiago Ramalho, Agnieszka Grabska-Barwinska, et al. Overcoming catastrophic forgetting in neural networks. Proceedings of the National Academy of Sciences, 114(13):3521–3526, 2017

2. Wang, Liyuan, Xingxing Zhang, Hang Su, and Jun Zhu. "A comprehensive survey of continual learning: theory, method and application." IEEE Transactions on Pattern Analysis and Machine Intelligence (2024).

**Questions:**

1. Why have state-of-the-art (non-replay based) approaches to alleviate catastrophic forgetting not been considered for comparison? For example, an approach well aligned with not having access to pre-training data is the Fisher-score based elastic-weight forgetting (EWC) technique [1] where the Fisher matrix is pre-computed and stored along with the pre-trained model weights.


2. How are $\tau$, $\alpha$, $\beta$ to be set / calibrated for fine-tuning? How are they tuned in the current setup?

## References:

1. James Kirkpatrick, Razvan Pascanu, Neil Rabinowitz, Joel Veness, Guillaume Desjardins, Andrei A Rusu, Kieran Milan, John
Quan, Tiago Ramalho, Agnieszka Grabska-Barwinska, et al. Overcoming catastrophic forgetting in neural networks. Proceedings of the National Academy of Sciences, 114(13):3521–3526, 2017

2. Wang, Liyuan, Xingxing Zhang, Hang Su, and Jun Zhu. "A comprehensive survey of continual learning: theory, method and application." IEEE Transactions on Pattern Analysis and Machine Intelligence (2024).

---

> ### Author Response · Authors · 2024-11-18
> **Author Response**
>
> Dear Reviewer G3zW,
>
> In response to Strength 1., where you refer to the results as “incomplete”: Please elaborate on the nature of the claimed incompleteness.
>
> Regarding the reported Weaknesses 1 and 2:
> The description of this weakness largely ignores the nature of this finetuning work, its embedding in the immediately relevant literature on efficient finetuning and low-resource methods, and appear to misclassify it as a continual learning effort. While we agree that similarities can be found, we again emphasize that the goal of this work is to *enable low-data finetuning*, and not to develop a new continual learning method for LLMs.
>
> Please revisit Section 6, especially related work of Li et al., 2020; Chronopoulou et al., 2021; Diao et al., 2023; Huang et al., 2023; Yang et al., 2024 to better understand the exact setting and methodology of the work. Please also observe the lack of comparisons to continual learning methods in these works.
>
> You further suggest implementation of metrics outlined in the review paper [2]. Please note that the metrics in the references sections are accuracy-based and used in multi-task learning scenarios, and thus are not applicable to the experimental methodology at hand (or in fine-tuning practice as a whole). We believe this reference has been included by mistake; we keenly await your correction on this matter to consider inclusion in our work.
>
> We await your response and stand available for further discussion.

---

> ### Comment · Reviewer_G3zW · 2024-11-24
> **Response to Authors**
>
> Dear Authors,
>     Thank you for your response. Here is a clarification of my reasoning behind citing paper [1] by Kirkpatrick et al. above.
>
> - **[No access to training data]** In [1], the authors propose an elastic weight forgetting (EWC) method which also doesn't assume access to the training data during fine-tuning. They only assume access to the `Fisher scores' for each parameter of the pre-trained model on the source dataset as a proxy for "importance" of the parameter in modeling the source task.  Hence it is different from replay-based techniques which you have employed (for context) in your paper while at the same time being more relevant to your paper as there is a common assumption of lack of access to training data during fine-tuning.
>
> - **[Specialization - Degeneralization Balance]** Separately, the `\lambda` hyperparameter in Eq. 3 of [1] can also be considered as balancing "specialization vs. degeneralization". This is another point of conceptual relevance because, as part of your first `contribution' in the paper, it is mentioned that MFT exhibits "markedly better trade-offs between forgetting of general domain and learning of specialized domain". Results in Fig. 3 further investigate the same concept making the comparison with [1] even more relevant.
>
> Hence, I request the inclusion of performance evaluation of the EWC technique mentioned in [1] into Table 2.

---

> ### Author Response · Authors · 2024-11-25
> **Further Author Response**
>
> Dear Reviewer G3zW,
>
> In response to your request, please find the results for EWC joint with the experiments requested by reviewer ZYc9 below.
>
> |              |        | PubMed |     |       | Pile of Law |     |       | OpenWebMath |     |       |
> | ------------ | ------ | ------ | --- | ----- | ----------- | --- | ----- | ----------- | --- | ----- |
> | Model        | Method | S      | DG  | Ratio | S           | DG  | Ratio | S           | DG  | Ratio |
> | Llama 3.1 8B | FT     | 7.0    | 1.4 | 0.20  | 6.4         | 1.5 | 0.23  | 2.6         | 0.9 | 0.35  |
> |              | LoRA   | 6.8    | 1.8 | 0.26  | 5.2         | 1.2 | 0.23  | 2.8         | 0.7 | 0.25  |
> |              | DoRA   | 6.9    | 1.6 | 0.23  | 5.0         | 1.1 | 0.22  | 2.8         | 0.9 | 0.26  |
> |              | IA3    | 1.2    | 0.1 | 0.08  | 1.5         | 0.0 | 0.01  | 0.6         | 0.0 | 0.08  |
> |              | MFT    | 6.6    | 0.9 | 0.14  | 5.1         | 0.4 | 0.08  | 2.3         | 0.4 | 0.17  |
> |              | Replay | 6.4    | 0.1 | 0.02  | 5.4         | 0.2 | 0.04  | 2.3         | 0.2 | 0.09  |
> |              | EWC    | 6.2    | 1.2 | 0.19  | 5.3         | 1.1 | 0.21  | 2.6         | 1.0 | 0.38  |
> | Llama 3.2 1B | FT     | 11.2   | 3.5 | 0.31  | 12.0        | 5.2 | 0.43  | 2.1         | 1.0 | 0.48  |
> |              | LoRA   | 10.8   | 3.8 | 0.35  | 9.9         | 5.0 | 0.51  | 2.0         | 0.9 | 0.45  |
> |              | DoRA   | 10.8   | 3.2 | 0.30  | 9.9         | 5.0 | 0.51  | 1.9         | 0.8 | 0.42  |
> |              | IA3    | 2.3    | 0.4 | 0.17  | 1.7         | 0.1 | 0.06  | 0.1         | 0.0 | 0.04  |
> |              | MFT    | 10.5   | 1.7 | 0.16  | 9.7         | 1.5 | 0.15  | 1.6         | 0.3 | 0.19  |
> |              | Replay | 7.5    | 1.5 | 0.20  | 11.3        | 1.0 | 0.09  | 2.1         | 0.0 | 0.02  |
> |              | EWC    | 7.9    | 2.6 | 0.33  | 11.0        | 4.9 | 0.45  | 2.3         | 1.0 | 0.43  |
> | Llama 3.2 3B | FT     | 9.1    | 1.6 | 0.18  | 8.3         | 2.8 | 0.34  | 4.0         | 1.9 | 0.48  |
> |              | LoRA   | 8.2    | 1.6 | 0.20  | 7.8         | 2.6 | 0.33  | 3.3         | 1.7 | 0.52  |
> |              | DoRA   | 8.2    | 1.5 | 0.18  | 7.8         | 2.6 | 0.33  | 3.3         | 1.7 | 0.52  |
> |              | IA3    | 2.1    | 0.2 | 0.10  | 1.7         | 0.1 | 0.06  | 0.3         | 0.1 | 0.33  |
> |              | MFT    | 7.6    | 0.9 | 0.12  | 7.5         | 0.8 | 0.11  | 3.1         | 0.4 | 0.13  |
> |              | Replay | 6.8    | 0.4 | 0.06  | 7.9         | 0.6 | 0.08  | 3.3         | 0.1 | 0.03  |
> |              | EWC    | 7.2    | 1.2 | 0.17  | 7.9         | 2.2 | 0.28  | 3.9         | 1.7 | 0.44  |
>
> >  In [1], the authors propose an elastic weight forgetting (EWC) method which also doesn't assume access to the training data during fine-tuning. They only assume access to the `Fisher scores' for each parameter of the pre-trained model on the source dataset as a proxy for "importance" of the parameter in modeling the source task. Hence it is different from replay-based techniques which you have employed (for context) in your paper while at the same time being more relevant to your paper as there is a common assumption of lack of access to training data during fine-tuning.
>
> Note that in order to compute the Fisher scores, one has to have access to the original data. Thus, the claim that "They only assume access to the `Fisher scores' for each parameter" is not accurate as the Fisher scores have to be computed somehow.
> Note also that the potential information capacity of the Fisher scores is non-negligible. For LLaMA 3.1 8B, the Fisher matrices computed take up about 16GB of memory, which roughly corresponds to half of the memory footprint of English Wikipedia text.
>
> In our case, we computed the Fisher scores on the control fraction of OpenWebText data (cf. Section 3.1."Process") and then loaded the Fisher matrices to the tuning on specialized domain. This is also why we list EWC under a separating line together with Replay.
>
> We hope that this satisfies your request. We continue to be available for discussion.

---

### Official Review · Reviewer_zZqU · 2024-11-04

**Soundness:** 3
**Presentation:** 4
**Contribution:** 4
**Rating:** 8
**Confidence:** 4

**Summary:**

The paper presents an algorithm for domain adaptation of large language models (LLMs), that mitigates catastrophic forgetting on the original pre-training tasks, without access to any of the pre-training data.

The method is a form of self-distillation, where pseudo-targets ("corrected" targets) are created by linearly combining the probability vector of the pre-trained model with the one-hot vector representing the new target, with a coefficient chosen so that the target token is the most probable one (by a margin), while minimizing the effect on the other tokens' predictions.

Extensive experiments on different families of models, and different sizes within a model family, show this method consistently gives a lower "degeneralization" rate (perplexity change on the original, general domain), and better generalization / degeneralization trade-off, compared to other methods that do not use data from the general domain (fine-tuning, different low-rank adapters).
Only the Replay method (which uses original training data) outperforms it.
Ablations are conducted to show that all the different mechanisms involved in this method are necessary to reach this performance.

**Strengths:**

Originality
--------------
1. To my knowledge, this is the first method specifically design to fine-tune off-the-shelf pretrained LLMs on a different domain, without having access to their training distribution.
2. The specific formulas derived for token-wise adaptation, including the special case of the original model's maximum prediction being already right, are not obvious.

Quality
----------
1. The various parts of the algorithm are well justified
2. Main experiments are comprehensive, across model types and sizes, and support the paper's conclusions
3. Additional experiments, like weeping over capacity of low-rank algorithms, and the margin parameter for MFT), further demonstrate the behavior of the metrics used (DG, DG/S) and algorithms compared
4. Incremental ablation showcase how the different parts of the algorithm proposed influence generalization and degeneralization.

Clarity
---------
The paper is extremely clear and well presented. Graphs give a more intuitive feeling of how the "correction" affects probabilities, and the derivation of coefficients in section 2 should be enough to re-implement the method.

Significance
----------------
A straightforward, reasonably efficient method to avoid over-fitting on a small SFT set is valuable, as LLMs adapted for different domain is an increasingly popular use case. The fact it is compatible with low-rank adapters (and can be combined with them) is also a point for its adoption.

**Weaknesses:**

Quality
----------
I understand why Replay had to be included as a baseline, despite its advantage, but I'm wondering if a variant of Replay using only synthetic data sampled from the pre-trained model could have made the point stronger that the actual training data is necessary, vs. Replay acting as a regularizer.

Minor points
-----------------
1. l. 222: Basline -> Baseline
2. l. 527: the citation for the Gemma paper shows up as "Team et al.", which is not really informative. Maybe update the bibtex to show "Gemma Team" or something.
3. Similarly, use braces in bibtex entries so that capitals are preserved, e.g., l. 609, `{OpenELM}` would avoid it being rendered as "Openelm". Same for "rlhf", "llm", "lora", etc.

**Questions:**

1. I'm wondering how a baseline closer to Replay, but using synthetic data sampled from the pre-trained model, would perform. I understand that it would be major additional experiments to perform, and I'm not asking for that, but I'm wondering if it's something you've considered addressing, or if there's literature of something similar being attempted. Intuitively, it could help further mitigating the degeneralization further away from the domain-specific distribution, by providing soft labels from a broader distribution, even if the synthetic data is not as good as the actual training data.

2. I'm also wondering what is the signification and importance of having $\tau$ in probability space, rather than in log-probability space. Is there a theoretical reason to have that margin expressed that way, or preliminary experiments showing it would be worse in log-prob?

3. Why start with further fine-tuning the pre-trained model on OpenWebText (l. 229)? What's the reason and implications of doing that, rather than take the off-the-shelf models directly?

4. l. 176 and below, is $p^T_{\textrm{argmax}\ p^T}$ just $\textrm{max}\ p^T$?

5. 2. In fig. 3, top right, it's weird to have the 1M line at 0 shortly before step 400, when it seems to have collapsed (Specialization hitting 0) in the top-left subfigure. I'd expect the DG/S ratio to be undefined, or arbitrarily high, rather than 0 at that point. Did I miss something?

---

> ### Author Response · Authors · 2024-11-18
> **Author Response**
>
> Dear Reviewier zZqU04,
>
> Thank you for your thorough review. In response to your replay point raised and question 1:
>
> > I understand why Replay had to be included as a baseline, despite its advantage, but I'm wondering if a variant of Replay using only synthetic data sampled from the pre-trained model could have made the point stronger that the actual training data is necessary, vs. Replay acting as a regularizer.
> The reason why we could not incl
> ude replay based on synthetic data is the inherent lack of distributional characteristics from the pre-training data when using such data. There is no “native” way to generate synthetic data that is representative of the pre-training distribution in terms of relative proportions of various samples, except perhaps drawing on a set of randomly sampled seed generation pretexts. And at that point, one is already very close to the standard replay.
>
> Regarding minor points, 1&3: thank you, implemented. Regarding point 2, this is unfortunately the doing of the ICLR template. The full citation (L640) starts with “Gemma Team” and continues with the list of additional authors.
>
> In response to your questions:
> 1. Please see above.
> 2. We stopped short of such optimizations. The target parameter $\tau$ was introduced as “the next simplest thing” when previous approaches considering fixed linear mixtures or naive direct corrections did not yield the performance we were seeking on small datasets. We suppose that the use of $\tau$ in the $p$-space is somewhat easier to interpret and comprehend, but concede that its introduction in the log-$p$ space could be more beneficial to optimization. That being said, we are currently already accomplishing all our goals in the low-data setting with the present approach.
> 3. We found that starting with very brief fine-tuning of the pre-trained model on OpenWebText helped to standardize the DG metric across different models. This helped with the large-vocabulary models (Gemma, Minitron) in particular, and we hypothesize that it is because these models owe some of their multi-lingual prowess to the size of their vocabularies. On the outset, we note that unifying several distinct families of models for a metric and on the data they have not been optimized remains to be a painstaking process.
> 4. Yes, that is correct. We retained the verbose indexing for consistency with other indexing, but yes, it is the maximum scalar.
> 5. In Fig 3 top-left (and therefore also top-right), the specialization line is actually heading for the negative region. This means that the model is overfitting on the small 1M-token dataset to the extent where more training on the training split actually results in even worse performance on the validation set than exhibited by the base model.  Observe that MFT successfully mitigates such undesirable behavior (cf. Fig 3 bottom-left).
>
> We stand available for further questions and discussion.

---

> > ### Comment · Reviewer_zZqU · 2024-11-26
> >
> > Thanks for the answers!
> >
> > I just have a follow-up question regarding synthetic data:
> > Is it not possible to sample from an empty prompt, with some non-zero temperature, and no beam search or anything? Wouldn't that be close to drawing from the implicit distribution of sequences? I'd expect that to work at least for decoder-only models, but I understand that might not be the case (maybe the post-training interfered with that ability?), but it would be nice to elaborate a bit on why that would not be feasible.
> >
> > Also, regarding the bibtex citation, I think you could use double braces in the .bib file, e.g., `{{Gemma Team}}`, so that it is treated as an organization name, not the name of a person ([source](https://tex.stackexchange.com/questions/149769/how-to-cite-organizations-name-with-space-between-words)).

---

> > > ### Author Response · Authors · 2024-11-27
> > > **Further Author Response**
> > >
> > > >  Is it not possible to sample from an empty prompt, with some non-zero temperature, and no beam search or anything? Wouldn't that be close to drawing from the implicit distribution of sequences? I'd expect that to work at least for decoder-only models, but I understand that might not be the case (maybe the post-training interfered with that ability?), but it would be nice to elaborate a bit on why that would not be feasible.
> > >
> > > The main issue with trying to retrieve the training distribution from a pre-trained model (which further assumes that the pre-training has not been followed by safety or some other finetuning) is that explorative sampling (i.e. attending to potential tokens with small probabilities) is prone produce nonsensical and non-grammatical text. From the perspective of the model, there is little difference between a statistically unlikely but grammatical and an non-grammatical continuation token. One is thus reduced to attending only to the most likely tokens, which then vastly reduces the space of possible texts.
> > >
> > > > Also, regarding the bibtex citation, I think you could use double braces in the .bib file, e.g., {{Gemma Team}}, so that it is treated as an organization name, not the name of a person (source).
> > >
> > > Thank you, we will follow your advice on the formatting of the citation for the rebuttal version of the manuscript.

---

### Author Response · Authors · 2024-11-27
**Rebuttal Revision Submitted**

Dear reviewers,

Thank you for taking the time to inspect the paper. Following the rebuttal discussions, we have incorporated your feedback into the manuscript, including results for most recent language models, results for comparisons to additional methods, better table captions, additional clarifications, and citation and typo fixes.

We continue to be available for discussion where needed.

---

### Meta-Review · Area_Chair_2Vn6 · 2024-12-23

**Metareview:**

This submission received a variety of positive comments from several reviewers, such as the method being identified as intuitive, straightforward, and easy to use by all reviewers. Reviewers also commented favourably on the extensiveness of experiments, which was further strengthened by the authors' additional empirical results provided as part of the rebuttal (such as experiments with EWC and more recent small SOTA models).

However, a serious common concern by almost all reviewers was the lack of clarity/delineation between the author's proposed setup and the more well-established field of Continual Learning (CL), both in goals and terminology. While the authors' argued that their submission ought to be considered a low-data fine-tuning method which shares only tangential similarities to CL, the fact that several reviewers were confused about this point shows that this argument is not convincing (A view I happen to share). Related, several reviewers commented on the lack of comparison to established Continual Learning methods to which the authors were unable to provide a full discussion and results beyond EWC (a method considered outdated for several years). Unfortunately, a simple reference to previously published work, which also did not attempt a more thorough comparison to the CL literature, is not convincing.

As a result, I regret to inform you that, in its current state, I do not recommend this paper for acceptance. I strongly recommend taking the comments regarding similarity to CL seriously and attempting a more direct comparison in the future. Several of the authors' comments on CL suggest that the authors are not yet sufficiently familiar with the CL literature to convincingly argue for the need of a new subfield/problem that arguably shares a lot of common ground with CL.

**Additional Comments On Reviewer Discussion:**

We saw a productive reviewer discussion, with authors and reviewers engaging in an exchange with each submitted feedback. The authors carried out a variety of additional experiments requested by reviewers, which resulted in a reviewer raising their score.

---

### Decision · Program_Chairs · 2025-01-22

Reject